



# Subglacial roughness of the Greenland Ice Sheet: relationship with contemporary ice velocity and geology

Michael A. Cooper[1], Thomas M. Jordan[1,2], Dustin M. Schroeder[2,3], Martin J. Siegert[4], Christopher N. Williams[1,5], and Jonathan L. Bamber[1]

[1]School of Geographical Sciences, University of Bristol, Bristol, UK.
[2]Department of Geophysics, Stanford University, Stanford, CA., USA.
[3]Department of Electrical Engineering, Stanford University, Stanford, CA., USA.
[4]Grantham Institute, and Department of Earth Science and Engineering, Imperial College London, London, UK.
[5]Now at British Geological Survey, Nottingham, UK.

*Correspondence to:* M. A. Cooper (m.a.cooper@bristol.ac.uk)

**Abstract.** The subglacial environment of the Greenland Ice Sheet (GrIS) is poorly constrained, both in its bulk properties, for example geology, presence of sediment, and of water, and interfacial conditions, such as roughness and bed rheology. There is, therefore, limited understanding of how spatially heterogeneous subglacial properties relate to ice-sheet motion. Here, via analysis of two decades worth of radio-echo sounding data, we present a new systematic analysis of subglacial roughness beneath the GrIS. We use two independent methods to quantify subglacial roughness: first, the variability of along-track topography—enabling an assessment of roughness anisotropy from pairs of orthogonal transects aligned perpendicular and parallel to ice flow; and second, from bed-echo scattering—enabling assessment of fine-scale bed characteristics. We establish the spatial distribution of subglacial roughness and quantify its relationship with ice flow speed and direction. Overall, the beds of fast-flowing regions are observed to be rougher than the slow-flowing interior. Topographic roughness exhibits an exponential scaling relationship with ice surface velocity parallel, but not perpendicular, to flow direction in fast-flowing regions, and the degree of anisotropy is correlated with ice surface speed. In many slow-flowing regions both roughness methods indicate spatially coherent regions of smooth bed, which, through combination with analyses of underlying geology, we conclude is likely due to the presence of a hard flat bed. Consequently, the study provides scope for a spatially variable hard bed/soft bed boundary constraint for ice-sheet models.

## 1 Introduction

The rate of global sea-level rise contributions from the Greenland Ice Sheet (GrIS) has accelerated over the past two decades (Velicogna and Wahr, 2006; Rignot et al., 2011). To constrain projections of future change, primarily driven by mass loss over the grounding line, models must parametrise characteristics influencing ice-sheet motion and dynamics (*e.g.*, Huybrechts, 1994;





Nick et al., 2013). Outlet regions, and in particular fast-flowing ice streams, are principally characterised by enhanced basal motion (basal sliding; Cuffey and Paterson, 2010; van der Veen, 2013). Conditions attributed to, and rates of, sliding at the bed are influenced by various properties of the subglacial environment, including, but not limited to: basal thermal regime; presence of basal water (and effective pressure); rheological bed properties (*i.e.*, presence of sediment and, its viscosity/deformability);

and, basal friction/traction (*i.e.*, resistance from bed roughness; Weertman, 1957; Nye, 1970; Durand et al., 2011; Clarke, 2004; Iverson and Zoet, 2015; Brondex et al., 2017; Stearns and van der Veen, 2018). Although the influence of these processes upon ice flow and dynamics are generally well understood (at least theoretically using idealized models; Cuffey and Paterson, 2010; van der Veen, 2013), they are not incorporated into ice-sheet models as spatially varying boundary conditions. Understanding the spatial variation in subglacial conditions and processes remains restricted by the paucity of observations; as such, necessary

model parameters are often inverted or inferred.

Fundamentally, ice-sheet models rely on the application of sliding laws to approximate the rate of basal-motion with regards to subglacial characteristics. Whilst several sliding laws exists, each variously influencing the behaviour and sensitivity of modelled glacier response (Brondex et al., 2017), most models rely on a Weertman-style hard-bed sliding law (Weertman, 1957, 1972; Stearns and van der Veen, 2018). In this case, sliding velocity, and thus broad characteristics of ice dynamics, are

controlled by frictional stresses induced at the ice-bed interface as a result of small-scale 'obstacles' (with a wavelength, or length scale, on the order of $\sim 1$ m) superimposed onto subglacial topography (with a length scale on the order of $\sim 100$–$1000$ m Weertman, 1957; Nye, 1970; Iverson and Zoet, 2015; Stearns and van der Veen, 2018). Such fine-scale obstacles are not resolved within widely available gridded bed topography products (*e.g.* Bedmap2, and BedMachine V3; Fretwell et al., 2013; Morlighem et al., 2017, respectively), and direct observation is not possible through conventional (*i.e.*, topographic) subglacial

roughness quantification methods utilising radio-echo sounding (RES) data (described below). Furthermore, the scale at which friction is induced by these features is much less than can be resolved within numerical ice-sheet modelling. As such, 'basal traction' is primarily simulated (inferred/inverted) using satellite-derived surface velocity (*e.g.*, Joughin et al., 2009; Durand et al., 2011; Arthern et al., 2015), with basal sliding inverted by optimally matching the model velocity to observations by reducing basal traction beneath specific regions of enhanced ice flow.

The quantification of subglacial roughness, and subsequent evaluation with regard to ice velocity, has been the focus of many studies in recent years across Antarctica (*e.g.*, Siegert et al., 2005; Rippin et al., 2006, 2014; Bingham and Siegert, 2007, 2009; Schroeder et al., 2014), and Greenland, though to a lesser extent (*e.g.*, Layberry and Bamber, 2001; Rippin, 2013; Lindbäck and Pettersson, 2015; Jordan et al., 2017). Whilst subglacial roughness appears to exert control on the location of fast-flowing streaming ice (Siegert et al., 2004; Rippin et al., 2006; Bingham and Siegert, 2007, 2009; Rippin et al., 2014),

the influence/behaviour with respect to ice motion is not universal. Existing roughness maps of Greenland (*i.e.*, Rippin, 2013; Jordan et al., 2017) show that fast-flow can be associated with rougher beds, where slow-flowing regions are more smooth. As the majority of studies to date quantify large-scale topographic roughness information (in the order of $\sim 1000$ m), any direct influence upon basal traction, if at all, remains unclear. However, a recent high-resolution assessment (sub-kilometre) of bed topography beneath Pine Island Glacier has concluded that small-scale bed features (order $\sim 10$–$100$ m) do indeed influ-





ence ice-motion, principally through the induction of basal drag controlled by the orientation and size of subglacial obstacles (Bingham et al., 2017).

Assessing subglacial roughness information with respect to ice motion, however, is not limited to basal traction, particularly when defined at varying length scales. When considering roughness signatures, Bingham and Siegert (2009) present a clear

conceptual framework for examining the causes and controls smooth- and rough-beds in both hard- and soft-bed situations. For example, the majority of roughness studies of the West Antarctic Ice Sheet bed have associated low roughness with the presence of deformable sediment (*e.g.*, Rippin et al., 2006, 2011, 2014; Bingham and Siegert, 2007); however, it is also evident that streamlined bedrock (hard-beds) promote smooth beds (*e.g.*, Siegert et al., 2005; Rippin et al., 2014; Jeofry et al., 2018). Altogether, this suggests not only that a consideration of orientation/anisotropy in the interpretation of subglacial roughness is

necessary, but also that basal motion relies on the influence of other factors (*e.g.*, basal thermal state, or geographical setting; Bingham and Siegert, 2009). Additionally, a recent characterisation of the majority of Greenland's outlet glaciers implies that the role of effective basal water pressure, and the availability of deformable sediment are more important and influential than basal friction itself (Stearns and van der Veen, 2018); however, it should be noted that this conclusion, and the role of friction in basal slip is contested (Minchew et al., 2019).

Conclusions drawn from previous quantifications of subglacial roughness in Greenland are limited. Whilst the broad, ice-sheet-wide distribution of roughness has been mapped (Layberry and Bamber, 2001; Rippin, 2013), systematic comparison to ice-motion, and in particular the relationship between roughness anisotropy and flow direction, has not been fully considered. Rippin (2013) presents the most recent, ice-sheet-wide depiction of subglacial roughness in Greenland. Whilst this highlighted the spatial distribution of roughness information across the island, a non-uniform conclusion was made with regard to ice

surface speed (ice surface velocity magnitude, $|v|$). Furthermore, the method employed aggregated information across various length scales, working to eliminate finer-scale information. More recently, Lindbäck and Pettersson (2015) present an (albeit spatially-limited) study highlighting the importance of considering roughness anisotropy, referenced to ice motion. The recent increase in coverage of RES data over the GrIS (Rodriguez-Morales et al., 2014; Morlighem et al., 2017), so far unused in roughness analysis, provides a new opportunity to increase understanding of the subglacial environment, enabling an ice-

sheet-wide description of spatially heterogeneous bulk (*i.e.*, geology, and presence of sediment) and interfacial properties (*i.e.*, roughness, and rheological bed properties).

Subglacial roughness information can be obtained from RES data in two ways. First, via the statistical properties of along-track topography (*e.g.*, Hubbard et al., 2000; Taylor et al., 2004; Siegert et al., 2005; Rippin, 2013; Goff et al., 2014; Jordan et al., 2017); and secondly, via the electromagnetic scattering properties of the bed-echo waveform (*e.g.*, Oswald and Gogineni,

2008; Schroeder et al., 2013; Young et al., 2016; Jordan et al., 2017). Topography-derived roughness can be obtained using both the space domain (*e.g.,* measuring the root mean square height as a function of horizontal length scale) and the frequency domain, or spectral methods (*e.g.*, performing a Fourier transform; Shepard et al., 1995; Hubbard et al., 2000; Shepard et al., 2001; Smith, 2014). The length scale over which topographic roughness is assessed is limited to be greater than the horizontal resolution of the RES measurements (typically 30 m or greater) (Taylor et al., 2004; Li et al., 2010; Jordan et al., 2017).

Scattering-derived roughness is sensitive to the radio wavelength in ice (typically 1–5 m for most radar systems), and reveals





more-fine scale geometric information about the subglacial interface than topographic analysis (Shepard et al., 2001; Berry, 1973; Schroeder et al., 2015; Jordan et al., 2017).

One simple approach to mapping subglacial information from electromagnetic scattering is to use the 'abruptness' (or 'pulse-peakiness') of the bed-echo waveform (Oswald and Gogineni, 2008, 2012; Young et al., 2016; Jordan et al., 2017). This parameter, defined as the ratio of peak to integrated bed-echo power, gives an indication of the relative contributions of specular reflection (higher abruptness and associated with fine-scale smooth beds) and diffuse scattering and clutter (lower abruptness and associated with fine-scale rough beds). RES flight-track maps for the bed-echo abruptness in northern and central Greenland demonstrate clear spatial structure (Oswald and Gogineni, 2008, 2012; Jordan et al., 2017). For example, there are near-continuous regions of high abruptness in the interior (*e.g.*, near the Camp Century and NorthGRIP ice cores; Oswald and Gogineni, 2008, 2012; Jordan et al., 2017), whereas many ice margin regions have lower abruptness levels (*e.g.*, the main trunk of Petermann Glacier; Jordan et al., 2017). The original geophysical interpretation of the larger-scale high abruptness regions (typically 100s of km$^2$) is that they often represent extended, electrically-deep ($> 8$ m; Gorman and Siegert, 1999), bodies of basal water (Oswald and Gogineni, 2008, 2012). However, this picture is largely inconsistent with ice-core temperature data and existing knowledge of the basal thermal state (MacGregor et al., 2016; Jordan et al., 2017). An alternative explanation is that the larger-scale high abruptness regions typically indicate smooth bedrock, with deep water only likely being present in localised patches (Jordan et al., 2017). This primarily lithological interpretation of the bed-echo abruptness has, however, yet to be fully explored and integrated with existing knowledge of ice dynamics and subglacial geology.

In this paper, using two decades worth of CReSIS RES data, we present a new systematic analysis for subglacial roughness beneath the Greenland Ice Sheet (GrIS). We outline two independent methods for quantifying roughness using information obtained via both statistical analysis of sampled bed elevation (hereafter termed, 'topographic roughness'; Sect. 2.2), and the scattering properties quantified from the bed-echo waveform (hereafter termed, 'scattering-derived roughness'; Sect. 2.3), respectively. We map the spatial distribution of subglacial roughness across the GrIS (Sect. 3), and document a marked spatial-heterogeneity using both metrics. We then assess roughness anisotropy (Sect. 3.2), providing clear evidence for direction-dependence between topographic roughness and the surface speed of ice in fast-flowing regions, both at the ice-sheet scale and locally, surrounding major outlet glaciers. Finally, to better understand the observed coherent signal of 'smooth' beds in regions of slow ice-flow we compare scattering-derived roughness to predicted underlying geology (Sect. 4.3).

## 2 Methods

### 2.1 Ice-penetrating radar systems and survey coverage

The RES data used in this study were collected by the Center for Remote Sensing of Ice Sheets (CReSIS) over the years 1993–2016, with more-recent campaigns undertaken as part of the wider Operation Ice Bridge (OIB) programme (post–2009). Surveys were typically undertaken between the months March and May, using three airborne platforms: a P-3B Orion (P3), a DHC-6 Twin Otter (TO), and a Douglas DC-8 (DC8) (Paden, 2017). The instruments used were, successively, the: Improved Coherent Radar Depth Sounder (ICORDS); ICORDS, version 2 (v2); Advanced Coherent Radar Depth Sounder (ACORDS);





Multi-Channel Radar Depth Sounder (MCRDS), Multi-Channel Coherent Radar Depth Sounder (MCoRDS), and MCoRDs, (v2) (Paden, 2017). Centre-frequencies for the radar instruments are 149 MHz (for ICORDS and ICORDS, v2), 150 MHz (for ACORDS and MCRDS) and 195 MHz (for MCoRDs and MCoRDS, v2). The vertical (depth-range) resolution varies from ~ 4.3 to 20 m, where the horizontal (along-track) resolution is typically ~30 to 60 m. Precise breakdown of the radar data

coverage by field season and radar instrument class can be found in MacGregor et al. (2015) (Fig. 1) and Jordan et al. (2018) (Fig. 1), respectively.

For measures of topographic roughness (Sect. 2.2) data across all campaigns were used; however, for scattering-derived roughness analysis (Sect. 2.3), only a subset of these are incorporated (indicated in Fig. 1), including ACORDS, MCRDS and MCoRDS, and MCoRDs v2 data. The rationale for this, relating to internal consistency when combining data from different

radar instruments, is described in Sect. 2.3. Additionally, owing to the preference for 'repeat fly-bys' in airborne sampling regimes, and the marked increase in survey kilometres in recent years (Rodriguez-Morales et al., 2014; Morlighem et al., 2017), the final spatial coverage of both roughness metrics is similar (Fig. 1).

Method-specific data pre-processing (*i.e.*, the handling of quality flags) is described below. For full information regarding the multiple radar instruments used in this analysis readers are referred to the user's guide (available from (http://data.cresis.ku.edu/data/rds/rds_

Paden, 2017). Additionally, detailed signal processing steps, and information regarding data segmentation, are described in several previous works (*i.e.*, Gogineni et al., 2001; Rodriguez-Morales et al., 2014; Gogineni et al., 2014; MacGregor et al., 2015; Paden, 2017).

## 2.2 Subglacial roughness from along-track topography

### 2.2.1 Calculating rms height, $R$

As noted, subglacial roughness information can be determined via the statistical analysis of vertical variation in along-track bed topography (*e.g.*, Siegert et al., 2004, 2005; Taylor et al., 2004; Rippin et al., 2006, 2011, 2014; Bingham and Siegert, 2007, 2009; Bingham et al., 2007, 2017; Li et al., 2010; Rippin, 2013). The most prevalent method in glaciological literature employs spectral methods to do this (*i.e.*, the application of fast Fourier transforms (FFTs) first employed in glaciology by Hubbard et al. (2000) and for the Antarctic Ice Sheet by Taylor et al. (2004)). Alternative space-domain methods exist, however, and are

frequently used within earth and planetary sciences (Shepard et al., 2001; Smith, 2014).

Here, the first metric for subglacial roughness we present, 'topographic roughness' (or $R$), is quantified by the root mean square (rms) height in along-track topography (RES sampled bed elevation). Rms height (referred to also as standard deviation of bed elevation; *e.g.*, Rippin et al., 2006, 2014) provides several benefits over the use of FFTs. First, it enables the collation of all CReSIS survey campaigns despite variable sample spacing (horizontal resolution) without requiring along-track interpola-

tion/ re-sampling data. Second, this method allows the use of a shorter length scale than FFT, not only facilitating subsequent anisotropic analysis at cross-overs (Sect. 2.2.2), but also providing a finer-scale roughness information. A final advantage is that rms height calculations are unit-preserving (*i.e.*, quantifying variation at the bed in units of metres), providing a more





physically-intuitive metric. More critically, however, the spatial distribution of roughness values quantified by FFT and rms height methods have been noted to be similar (Rippin et al., 2014; Falcini et al., 2018).

Sampled bed and surface elevations were obtained from all the available CReSIS RES surveys (1993–2016). Where applicable, data were filtered using the provided quality flags denoting the confidence of the bed pick accuracy (Paden, 2017),

ensuring only bed elevations with 'high' confidence were used; however, as RES data obtained during OIB campaigns prior to 2008, with the exception of the reprocessed '2006 TO' survey, do not include quality flags, all available sampled bed elevations were used. As a result of the increased sampling resolution in more recent surveys (post–2006), data for these campaigns were rarefied (to include every other sample point), to ensure only independent measures of bed elevation were used (Paden, 2016, pers. comm.).

Topographic roughness, $R$, is given by

$$R = \left[ \frac{1}{n-1} \sum_{i=1}^{n} (z(x_i) - \overline{z})^2 \right]^{\frac{1}{2}}, \tag{1}$$

where $n$ is the number of sample points, $z(x_i)$ is the height of the surface point at point $x_i$, and $\overline{z}$ is the mean height of the profile over all $x_i$. Rms height $R$ was calculated using a window length or bin size, $L$, of 200 m using all recorded bed elevations, regardless of spatial density within the bin, provided $n \geq 3$. $R$ is given for the spatial midpoint of each window.

Regions of greater roughness, quantified by a larger variation in bed elevation within the window, have greater $R$ values. An example of $R$ calculated along-track using sampled bed elevation is presented in Fig. 2.

$L = 200$ m was chosen to enable the finest-scale of $R$ to be quantified whilst maintaining the largest spatial coverage of the resultant metric by using all available survey data. It is possible to quantify $R$ at a finer scale using only more recent survey data, however this is at the expense of reduced spatial coverage, for a length scale not less than 100 m. It should be noted that

not all bins have constant $n$, due to the variation in sampling regime, the resolution of the different radar instruments, and data quality. $R$ was not calculated for bins where $n < 3$. Whilst $n$ is small (mean $= \sim 8$) in the roughness statistics used in this paper we obtain a large sample size through the repeated sampling over multiple bins (repeat flight tracks). This repeated sampling approach for small $n$ parallels how statistically robust estimates are made when calculating the scale-dependence of roughness using a variogram (Shepard et al., 2001).

**2.2.2  Filtering $R$ with respect to ice surface velocity**

To evaluate, and more completely understand, how the spatial distribution of subglacial roughness influences, or perhaps is influenced by ice-sheet motion, we compare $R$ to ice surface velocities. We use the InSAR-derived MEaSUREs velocity mosaic (Joughin et al., 2016, 2017) over the entire GrIS. This mosaic helps to capture long-term information (using 1995–2015 observations) regarding flow configuration, minimising inter- and intra-annual variation in both ice speed and direction.

As $R$ is quantified using two decades worth of RES data, we assume an inherent constancy in roughness over time. The MEaSUREs data provides magnitude ($|v|$; speed) and direction at a 250 m resolution (Figs. 3(a) & (b)); however, for our analysis, we performed a bilinear aggregation (to 1000 m) in order to smooth small-scale variation or noise.





With regard to ice-surface flow speed we delineate regions of 'fast' ($|v| \geq 50$ ma$^{-1}$) and 'slow' ($|v| \leq 5$ ma$^{-1}$) flow (Fig. 3 (a)). In regions where $|v|$ exceeds 50 ma$^{-1}$, ice is likely to be decoupled from the bed (*i.e.*, sliding) as this speed cannot be achieved by internal deformation alone (MacGregor et al., 2016; Stearns and van der Veen, 2018). As we have noted above, basal traction, a principal constraint on basal sliding (Weertman, 1957), may be influenced by subglacial roughness (Siegert et al., 2004, 2005; Bingham et al., 2017). Second, where ice motion is limited in slow-flowing regions, rates of basal erosion are minimal and, thus, the influence on subglacial topography is reduced (Bingham and Siegert, 2009). It should be noted that we only use contemporary ice velocity observations in this study; although flow configuration is likely to have remained largely constant, the surface speed will have changed through time.

As $R$ is quantified along-track, there is an inherent directionality in its characterisation of the subglacial environment. To assess anisotropy at the bed, with particular reference to ice-motion, we classify $R$ through its alignment with local flow direction. Sample windows were filtered for their linearity to remove measures of $R$ over corners and bends (with a deviation $\geq 10\%$; after Bingham et al., 2015) in RES flight-lines/transects. Roughness bins were then filtered by their alignment to local surface ice flow direction (Fig. 3 (b)) with a 20° threshold; Fig. 3 (c) shows classified measures of $R$ aligned perpendicular $R_\perp$ or parallel $R_\parallel$. From this, we draw conclusions based on the relationship between subglacial roughness and the speed of overlying ice.

Where coincident measures of $R_\perp$ and $R_\parallel$ are available (the near-orthogonal ($\pm 20°$) cross-overs between flight-lines) the degree of anisotropy can be calculated. This is achieved through a normalised difference ratio, herein termed 'anisotropy ratio' (Smith, 2014), given by

$$\Omega = \frac{R_\parallel - R_\perp}{R_\parallel + R_\perp}. \tag{2}$$

Here, using $\Omega$, we map the distribution of roughness anisotropy across the GrIS and assess the relationship between $|v|$ and $\Omega$ in both fast- and slow-flowing regions. Values of $\Omega$ are interpreted such that $-1$ dictates a complete dominance of smoothness parallel to flow direction (perhaps as a result of flow-aligned features), where $+1$ a dominance of smoothness perpendicular to flow (*i.e.*, parallel roughness), and values of $\sim 0$ indicates roughness isotropy.

## 2.3 Subglacial roughness from radar scattering

### 2.3.1 The abruptness (peakiness) of the bed-echo waveform

Bed-echo waveform properties are related to electromagnetic scattering from the glacier bed and, hence, also provide information about subglacial roughness (Oswald and Gogineni, 2008; Oswald et al., 2018; Jordan et al., 2017). Radar bed-echoes range from sharp pulse-like returns (associated with specular reflections from a smooth glacier bed), to echoes that have a trailing edge that extends greatly over the original pulse length (associated with diffuse scattering from a rough glacier bed). A convenient way to parametrise the relative spread of the bed-echo waveform is to use the waveform 'abruptness' parameter defined by

$$A = \frac{P_{peak}}{P_{agg}}, \tag{3}$$





where $P_{peak}$ is the peak power of the bed-echo and $P_{agg}$ is the aggregated (integrated) power over the echo envelope (Oswald and Gogineni, 2008; Jordan et al., 2017). Three examples of bed-echo waveforms, and their abruptness values, are shown in Fig. 4(c). Higher values of $A$ are associated with specular reflections, and lower values with diffuse scattering. However, the maximum value for $A$ (which is constrained by the ratio of the image sample rate to depth-range (vertical) resolution/

bandwidth (Jordan et al., 2017)) can differ between different CreSIS field seasons with values ranging between 0.5 and 0.8. Since the RES bed-echo results from a superposition of along-track and cross-track energy, the abruptness is a (near) isotropic parameter (Young et al., 2016), and therefore obscures information regarding the anisotropy of the glacier bed.

The procedure used to extract the bed-echo abruptness from CReSIS Level 1B data is outlined in Jordan et al. (2017). Briefly, this consists of the following three steps. First, CReSIS Level 2 picks are used as initial estimates for the depth-range bin of

bed-echo power peak. Second, a local re-tracker is used to locate peak-power. Third, the power is integrated over the bed-echo envelope applying a 'quality control' measure such that the peak power is 10 dB over the noise floor. This final step results in some regions, primarily in Southern Greenland, having reduced coverage (see Fig. 1(b) in Jordan et al., 2018)).

### 2.3.2    Estimating fine-scale roughness and the 'peakiness index'

The scattering of the radar pulse at the glacier bed is underpinned by the physics of electromagnetic diffraction (Berry, 1973;

Ulaby et al., 1982). As bed roughness increases, the radar pulse is scattered over a greater range of angles; this results in a decrease in peak returned-power, and an increase in the trailing edge of the echo. The mathematical formulation of this relationship depends on the physical model for electromagnetic interference (phase coherence, or incoherence) and the statistical model for the subglacial interface (Berry, 1973; Peters et al., 2005; Haynes et al., 2018). The most commonly employed scattering model for the RES of glacier beds assumes phase-coherent interference, 'smoothly undulating' Gaussian statistics for

rms roughness and radial isotropy (Berry, 1975; Peters et al., 2005; MacGregor et al., 2013; Grima et al., 2014; Schroeder et al., 2015). We employ this scattering model for two objectives: firstly, as a way of estimating 'fine-scale roughness' from the abruptness; and, secondly as a way of combining the abruptness for different radar systems to derive an (approximately) system-independent 'peakiness index' ($\Lambda$).

Following a similar approach to that described by Schroeder et al. (2015) and Jordan et al. (2017), under assumptions of

energy conservation, the scattering model can be use to predict the relationship between $A$ and rms height $\xi$ ('fine-scale' roughness). In this context $\xi$ is not strictly equivalent to the values obtained from topography, and a length scale separation is performed with respect to a reference plane (Berry, 1973). The relationship between $A$ and $\xi$ is given by

$$A = A_{max}\exp(-g^2)I_0^2\left(\frac{g^2}{2}\right), \tag{4}$$

where

$$g = 4\pi\xi f_c\sqrt{\epsilon_{ice}}/c, \tag{5}$$

denotes the rms phase variation, with $A_{max}$ the maximum abruptness, $I_0$ a zeroth-order Bessel function of the first kind, $f_c$ is the centre-frequency of the radar pulse, $c$ is the vacuum speed of the radar pulse and $\epsilon_{ice} = 3.15$ is the relative dielectric





permittivity of glacier ice (Peters et al., 2005). Since the radar wavelength in ice is $\lambda_{ice} = c/f_c\sqrt{\epsilon_{ice}}$, eq. (5) can be expressed as

$$g = 4\pi\xi/\lambda_{ice}, \tag{6}$$

and hence $\xi$ is scaled by the radar wavelength in ice (either 0.87 m or 1.13 m for the 195 MHz and 150 MHz systems,

respectively). There are therefore two degrees of freedom in eq. (4) that can vary for different CReSIS field seasons: $A_{max}$ and $f_c$. The different parameter combinations are shown in Fig. 4(a), and from these relationships it is possible to estimate $\xi$ from $A$ (and thus obtain a measure of fine-scale roughness that is similar between different radar systems). However, since the values of $f_c$ and $A_{max}$ differ between field seasons a cross-over bias is present for 'raw' abruptness values. In order to combine abruptness data we back-substituted the value of $\xi$ to obtain the value of $A$ as if it were the most spatially extensive

radar system (the blue curve in Fig. 4(a)), and then re-scaled amplitude on the interval $[0,1]$ to give the 'peakiness index' (from herein referred to as $\Lambda$). These steps combine the measurements via the system-independent relationship that is modelled between $\Lambda$ and wavelength-scaled rms height $\xi/\lambda$ (Fig. 4(b)).

The inter-season data combination was validated by performing cross-over analysis for $\xi$ and $\Lambda$, with the allowed tolerance for the cross-over bias set to $5\%$ of the parameter range. RES data that do not meet this criterion (primarily the older ICORDS

data, but also the 2010 P3 season which is known to have noise-floor issues Paden (2017)) were discounted completely from analysis. Although the data combination scheme employed here, across CReSIS platforms, is seen to work well, it should be noted that combining data from multiple instruments, particularly those with a large difference in center frequencies, may not be so effective.

It is important to note that obtaining $\xi$ from eq. (4) is just one way of estimating fine-scale roughness. Self-affine (fractal)

statistics (Shepard and Campbell, 1999) can also be applied to scattering models of glacier beds (as in Jordan et al., 2017). Additionally, in reality, fine-scale roughness is anisotropic as revealed by the 'specularity' scattering metric (Schroeder et al., 2013, 2014, 2015; Young et al., 2016). We therefore recommend that $\xi$ should be interpreted in a qualitative manner, with lower values indicating 'fine-scale smooth' and higher values indicating 'fine-scale rough' regions of the glacier bed. In regions of complex bed topography, and in particular at outlet glacier regions, off-nadir scattering may adversely influence the signal and

lead to a breakdown in the interpretation of this metric (see Sect. 4.4). Fine-scale roughness that relates to radar scattering can also be estimated from the statistical distribution in peak bed-echo power (Neal, 1982; Grima et al., 2014).

## 3   Results

### 3.1   Spatial distributions for subglacial roughness

#### 3.1.1   Topographic roughness, $R$

Across the ice-sheet, unfiltered (with respect to surface flow direction) $R$ shows clear spatial-heterogeneity (Fig. 5(a)); coherent signals, representing contiguous regions of both 'smooth' (low $R$ values) and 'rough' (high $R$ values) beds, are visible.



Generally, the margins of the ice sheet contain the roughest beds, whereas the interior is notably smooth. Ice-sheet-wide, the lowest values of $R$ are observed in the north and north-west of the island. However, localised to the main 'trunks' of Petermann and Humboldt Glaciers, at the point of highest $|v|$ immediately before the grounding line, small patches of smooth bed are observed. Broadly speaking, across the ice-sheet, fast-flowing regions exhibit rough beds, though, as exemplified in the north

and north-west, this behaviour is somewhat spatially-variable at the perimeter of Greenland. Notable examples of contiguous smooth beds near the margins include: northwest of the Camp Century (CC) drilling site; in the vicinity of Ìngia Isbræ (II; north of Rink Isbræ); and, a region near the outlet of the North East Greenland Ice Stream (NEGIS) (as marked on Fig. 5(a)). The highest values of $R$ trace the Caledonian fold belt mountain range (formed $\sim$420 Ma B.P.; Henriksen, 2008) and the deep inland fjord-like systems along the east and south-eastern margins of the island (Fig. 5(a)).

Figures 5 (b) & (c) present directionally-filtered values for topographic roughness, aligned perpendicular ($R_\perp$) and parallel ($R_\parallel$) to ice surface flow direction, respectively. For improved visualisation, maps for $R_\perp$ and $R_\parallel$ were interpolated (using inverse-distance weighting) to a limit of 10 km (Figs. 5 (d) & (e)). This interpolation distance is representative of the average track-spacing used in the 'gridded' airborne sampling regimes in fast-flowing regions (*e.g.,* surrounding Jakobshavn Isbræ and Petermann Glacier; see Fig. 1). Initial comparison shows a marked difference between $R_\perp$ and $R_\parallel$, most notably within fast-

flowing regions. Across the ice-sheet, the bed is observed to be smoother parallel to flow. In the ice-sheet interior (where $|v| <$ $50\ \mathrm{ma}^{-1}$) the subglacial environment is mostly smooth in both directions (*i.e.*, isotropic). However, in the south of the ice sheet we observe more distinct differences between $R_\perp$ and $R_\parallel$ values (see Sect. 3.2). Overall, $R_\parallel$ exhibits more uniform roughness values across fast- and slow-flowing regions, particularly within the north and west, whereas $R_\perp$ presents a notable difference between fast- (rough) and slow-flowing regions (smooth).

We observe a similar spatial distribution of unfiltered $R$ (Fig. 5(a)) to those previously quantified for Greenland using an rms residual technique (see Fig. 4 in Layberry and Bamber, 2001), and through a frequency-domain approach (FFTs) undertaken at a much larger length scale (3,200 km; see Fig. 1 in Rippin, 2013); in these studies, general conclusions for a smooth interior and rough margin were made. Rippin (2013) additionally note a localised smooth bed underlying the trunk of Petermann Glacier, whereas Layberry and Bamber (2001) notes a smooth basin for both Humboldt and Petermann glaciers. However, as these

studies do not filter with respect to surface flow direction, they do not reveal roughness anisotropy in $R$ across the ice-sheet.

### 3.1.2   Scattering-derived roughness, $\xi$

Figure 6(a) presents the spatial distribution of scattering-derived subglacial roughness, $\xi$, for the GrIS. As noted (Sect. 2.3), these values are inversely correlated to $\Lambda$ (Fig. 6(b)), due to the scattering model relationship. The spatial distributions observed within scattering-derived roughness are broadly similar to that observed for unfiltered $R$, including a notable link between

fast-flow and high values of $\xi$ (rougher beds). Regions that present the smoothest subglacial environments also reflect those mentioned above, notably: the vicinity of the CC drilling site; a coherent patch south-east of Petermann Glacier; towards the outlet of the NEGIS; and, at Ìngia Isbræ (marked on Fig. 6(a)). Low (smooth) values of $\xi$ are also observed along the central ice divide. Contrasting to measures of $R$, however, and concordant with the broad-scale relationship of $\xi$ to $|v|$, the fastest-flowing trunks of Humboldt and Petermann Glaciers contain rougher beds. Other differences between topographic and





scattering-derived roughness include a corridor of high $\xi$ extending south of Petermann Glacier and across ice divide (see Fig. 6), as well as a generally more 'mixed' roughness behaviour in the ice-sheet interior.

## 3.2 Relationship with contemporary ice velocity

### 3.2.1 Ice-sheet scale

Owing to the isotropic nature of $\xi$, we limit more comprehensive assessment of the relationship between contemporary ice velocity and subglacial roughness to topographic roughness, $R$, only. Figure 7 presents an assessment of the relationship between $R$ with respect to surface ice-flow direction in fast-flowing regions ($|v| > 50$ ma$^{-1}$). The difference in distributions between $R_\perp$ and $R_\parallel$ (Figs. 7(a) & (b)) indicates that roughness perpendicular to flow direction is greater (*i.e.*, more 'rough;' mean = 9.39 m, compared to 6.27 m) and exhibits higher variance (92.21 m$^2$, compared to 43.02 m$^2$).

Calculated mean ice surface speed ($|\bar{v}|$) for logarithmic bins (at 0.25 intervals) of $R_\perp$ and $R_\parallel$ are shown in Figs. 7(c) and (d), respectively. A marked difference between the calculated ice speed averages is observed. For all bins of $R_\perp$, $|\bar{v}|$ is seen not to exceed 250 ma$^{-1}$, whereas the lower bound for $|\bar{v}|$, calculated for $R_\parallel$, is $> 350$ ma$^{-1}$. This scaling relationship is broadly in agreement with those previously observed in the literature for Antarctica (Bingham and Siegert, 2007) and Greenland (Lindbäck and Pettersson, 2015); however, it is notable that this relationship is evident for the ice sheet as a whole, compared

to these regional studies. Additionally, if we are to assume that $|v|$ increases toward the glacier terminus/grounding line, the exhibited scaling relationship for $R_\parallel$ is in agreement with previous studies where roughness is observed to decrease (Bingham and Siegert, 2007, 2009). Increasingly smooth beds parallel to flow direction, therefore, are indicative of enhanced ice surface speed. The limit to which this relationship holds is $R = 10^{1.25}$ (also delineated in distribution histograms by the dashed black line in Figs. 7 (a) and (b)). This value is the approximate upper limit of $R$ that can reasonably quantified using eq. 1 (Sect. 4.4).

Conversely, a weak positive relationship is observed between $R_\perp$ and mean ice surface speed (Fig. 7(c)). $R_\parallel$, however, exhibits a strong negative exponential scaling relationship with mean ice surface speed (Fig. 7(d)), which is statistically significant above the $p = 0.001$ confidence level.

Figure 8 (a) presents the spatial relationship of the anisotropy ratio ($\Omega$) across the ice-sheet, where coincident values of $R_\perp$ and $R_\parallel$ are quantified. It is clear that fast-flowing outlet regions (the ice-sheet margins) are generally more smooth parallel to

surface flow direction (where $\Omega \to -1$). In the ice-sheet interior a more varied/random distribution in $\Omega$ is apparent. Mean ice surface speed for bins of $\Omega$, at 0.1 intervals, in fast- and slow-flowing regions (Figs. 8 (b) & (c)), reinforces this observed spatial relationship in subglacial roughness. A strong linear relationship with regards to $|v|$ is exhibited within fast-flowing regions, whereas in regions of slow-flow no such relationship is observed.

### 3.2.2 Fast-flow regions and outlet glaciers

To assess any spatial-heterogeneity in the exponential scaling relationship between ice flow and $R_\perp$, local regions of fast-flow were selected for closer analysis. These regions are centred around major outlet glaciers (Fig. 9) and, where possible, encompass only individual outlet glaciers (*e.g.*, Humboldt [Region 1], Petermann [2], and Kangerdlugssuuaq [4]); however, where





outlet glaciers are in close proximity, wider regions of fast-flow were assessed (*i.e.*, 'Jakobshavn+' [Region 6]). Regionally, we observe the same exponential scaling relationship as exhibited ice-sheet-wide. The calculated regression line for each region is statistically significant at, or above, the $p = 0.01$ confidence level, with the exception of Region 3 (encompassing NEGIS) at $p = 0.05$. A marked difference in the regression gradients is also observed, spanning four orders of magnitude: Region 3 ex-

hibits the shallowest gradient ($-1.01 \times 10^{-1}$ a$^{-1}$), and Regions 4 & 5 the steepest ($-9.39 \times 10^{-4}$ a$^{-1}$ and $-7.66 \times 10^{-4}$ a$^{-1}$, respectively). Echoed by the shallow regression gradient and the lower confidence level of statistical significance, the NEGIS (Region 3) also exhibits the lowest r-squared value (0.35). As previously descrived, both unfiltered $R$ and $\xi$ values reveal a contiguous smooth bed signal, aligned near-perpendicular to flow direction (marked on Figs. 5(a) & 6(a); further described in Sect. 4.3). Downstream from this, a coincident increase in subglacial roughness and $|v|$ is observed. Additionally, there is a

notable sampling bias in the radar sounding across Region 3, where fewer tracks are aligned parallel to the flow direction (Figs. 3(c) & 9(b)). Together, these factors may be responsible for the weaker scaling relationship observed here between $|\bar{v}|$ and $R$.

More interestingly, two distinct groups are observed, showing a clear separation in regression slope gradients (Fig. 10). The first group (see bottom; Fig. 10) are mostly-homogenous in terms of their regression slopes (*i.e.*, the relationship between roughness and ice surface speed here is broadly similar). However, Regions 4 & 5 in south-east Greenland (Kangerdlugsuuaq

and Helheim, respectively) exhibit marked increase in gradient, indicative of a stronger scaling relationship at these sites.

### 3.3 Contiguous smooth beds in slow-flow regions

To recap, we observe coherent, contiguous 'smooth' regions present across the GrIS across both roughness metrics (Figs. 5 & 6). These regions include north-west Greenland (around CC; Fig. 11); south-east of Petermann Glacier (Fig. 12); and bisecting central Greenland bounded west–east by Ìngia Isbræ and Geikie Plateau, respectively (II and GP; Fig. 13). Owing to

its isotropic nature, and inherent sensitivity to more fine-scale roughness information, we have focused on measures of $\xi$ for these regions. High abruptness values (comparable to $\Lambda$; Fig. 4(b)) in several of these regions has previously been observed (*e.g.*, Fig.6(c) in Jordan et al., 2017; Oswald and Gogineni, 2012). For the most part, these are coincident with regionally high, and flat, beds (Morlighem et al., 2017), slow surface ice speed (Joughin et al., 2016) and a frozen basal thermal state (MacGregor et al., 2016; Jordan et al., 2017).

### 4 Discussion

#### 4.1 Interpretation of spatial patterns

As previously mentioned, Weertman-style hard-bed sliding laws are theoretically influenced/limited by basal traction exerted on the ice column by small-scale basal obstacles (on the order $\sim 1$ m) (Weertman, 1957; Nye, 1970; Durand et al., 2011). However, the most prevalent methods of quantifying subglacial roughness (*i.e.*, through statistical analysis of along-track bed

elevation, as in this study) are limited to evaluating basal information directly at the order of 100–1000 m, or downscaled using fractal parameters (as in Jordan et al., 2017). Nevertheless, in regional studies of West Antarctica (*e.g.*, the Siple Coast), a





smooth bed has widely been considered a control on the location of fast-flowing, streaming ice (Siegert et al., 2004; Bingham and Siegert, 2009) and, in contrast, slow-flowing regions have been observed to widely exhibit more-rough beds (Siegert et al., 2004; Bingham and Siegert, 2007; Rippin et al., 2006, 2014).

However, when assessed across Greenland, it is evident that the spatial relationship between subglacial roughness and $|v|$ appears to be non-universal (in particular, fast-flowing regions can be both rough and smooth). In direct contrast, rough beds have been observed coincident with contemporary fast-flowing ice both in Antarctica (Schroeder et al., 2014; Bingham et al., 2017), and previously in Greenland (Rippin, 2013; Jordan et al., 2017). In this study, as exhibited across both unfiltered topographic roughness ($R$) and the more-fine scale, scattering-derived roughness ($\xi$) measure, a similar spatial relationship to $|v|$ is observed (Figs. 5(a) & 6(a)). Rough beds are seen to dominate fast-flowing regions, where slow-flowing regions are

predominantly smooth. This relationship, therefore, does not fit within a classical interpretation of roughness influencing basal traction, nor does it suggest that smooth beds are a necessary condition for fast-flow across Greenland. This finding is in broad agreement with a recent evaluation of basal motion across Greenland's outlet glaciers, whereby basal traction is concluded not to be controlled by Weertman-style hard-bed sliding law, but rather is influenced by soft beds and/or the presence of basal water (the Zwally effect) (Schoof, 2010; Moon et al., 2014; Chu et al., 2016; Stearns and van der Veen, 2018). Further interpretation

of the relationship between subglacial roughness, namely flow-filtered topographic roughness ($R_\perp$ & $R_\parallel$), and $|v|$ is given below (Sect. 4.2).

Where a direct influence upon basal traction is elusive, the interpretation of subglacial roughness has been centred on geomorphic means. One such framework is outlined by Bingham and Siegert (2009), whereby smooth-bedded regions have been associated with the presence of deformable sediment, perhaps attributable to marine sedimentation (*e.g.*, Rippin et al.,

2006, 2011, 2014; Bingham and Siegert, 2007), or as a result of enhanced erosion resulting in topographic streamlining within bedrock (*e.g.*, Siegert et al., 2005; Rippin et al., 2014). Low-lying topographic basins, particularly within a marine setting, may promote a smooth-bed owing to marine deposition/sedimentation during deglaciated periods (Bingham and Siegert, 2009). In this vein, the localised, relatively-smooth bed observed underlying NEGIS may be a likely candidate for deformable sediment (marked, Fig. 5(a) & 6(a)). Christianson et al. (2014) characterises the presence of subglacial till in this region through seismic

analysis; this is coincident with a marine-overdeepening underlying NEGIS as well as low $R$ and $\xi$ values (smooth beds) as quantified in this study (Figs. 5(a) & 6(a)). More in-depth assessment of the presence of sediment, alongside the evaluation of hard (non-deformable) beds, is further discussed below (Sect. 4.3).

Much of the ice-sheet interior is characterised by a frozen basal thermal state (MacGregor et al., 2016), which, alongside low $|v|$, suggests that rates of erosion or sediment transport (deposition) is negligible. Smooth beds in regions slow-flow, have

previously been characterised as markers of palaeo-ice streams, or fast-flow, in regional Antarctic studies (*e.g.*, Siegert et al., 2005; Bingham and Siegert, 2009; Lindbäck and Pettersson, 2015). Whilst such an interpretation of the smooth-bedded interior across Greenland (Figs. 5(a) & 6(a)) is not feasible for the modern ice sheet, it may be plausible to attribute this to the waxing and waning of the GrIS over multiple interglacial cycles.



## 4.2 Interpretation of roughness-velocity scaling relationships

As noted above, the consideration of orientation within subglacial roughness interpretation is important (Gudlaugsson et al., 2013; Falcini et al., 2018), despite previously being limited to regional studies (*e.g.*, Bingham and Siegert, 2007; Lindbäck and Pettersson, 2015). Analysis of flow-filtered $R$ values demonstrates a pronounced anisotropy of the subglacial roughness. Not

only is this observed ice-sheet-wide at crossover measures via the anisotropy ratio ($\Omega$; Fig. 8), but also in the marked difference in roughness behaviour in fast-flowing regions ($|v| > 50$ ma$^{-1}$; Fig. 7). Distributions of $R_\perp$ and $R_\parallel$ values suggest that the subglacial environment of Greenland is not only more smooth aligned parallel to flow direction on average, but that $R_\parallel$ tends towards smaller values (Figs. 7(a) & (b)), giving rise to different relationships between $|\bar{v}|$ and $R_\perp$ and $R_\parallel$ (Figs. 7(c) & (d), respectively).

Where the length scale of $R$ is too great to directly relate to basal traction within a Weertman-style hard-bed sliding law (Weertman, 1957; Nye, 1970), and the low-likelihood of such a control on ice motion (Stearns and van der Veen, 2018), a different interpretation must be made with reference to the exhibited roughness-velocity scaling relationships. As such, increasing $|v|$ is unlikely to be explained by an decrease in $R_\parallel$ values; this change is more likely attributable to enhanced erosion or sediment transport (increasing with $|v|$), resulting in a streamlining/elongation of bed features, possibly within deformable

sediment (*e.g.*, mega-scale glacial lineations (MSGLs) observed in King et al., 2009; Schroeder et al., 2014; Bingham et al., 2017). Additionally, the, albeit weak, positive relationship between $|\bar{v}|$ and $R_\perp$ could be plausibly explained by enhanced erosion increasing cross-feature amplitude (greater $R_\perp$ values) of streamlined beds. Generally, the spatial distribution of $R_\perp$ values present a more marked difference between fast- and slow-flow regions, when compared to values of $R_\parallel$. This is most likely influenced by velocity-controlled bed morphology, including both large-scale troughs/valleys linear bedforms, such as

MSGLs.

The roughness-velocity scaling relationship observed parallel to the flow direction is seen to be locally-variable (Fig. 10). The likely cause for the clear separation, or 'grouping,' within the regression gradients is likely due to the nature of the underlying topography. Kangerdlugsuuaq (Region 5) and Helheim (4) glaciers are classically defined as being 'topographically-constrained,' by which flow is steered to the margin through steep-sided valleys/troughs. This influences the onset of flank flow,

providing more lateral control to fast-flowing ice and its basal motion, impacting upon local rates of erosion and/or deposition. Although Jakobshavn Isbræ is also considered to be topographically constrained, we do not see such a pronounced relationship for the 'Jakobshavn+' region (Region 6; Fig. 10). This is likely because we have conglomerated neighbouring glaciers together due to their spatial density; however, this does suggest that topography provides less lateral control in this region, as remarked by Rippin (2013).

## 30 4.3 Interpreting hard bed geology

In fast-flowing regions ($|v| > 50$ ma$^{-1}$), we observe mixed behaviour in subglacial roughness. Parallel to ice flow direction ($R_\parallel$), smooth beds are a likely a result of enhanced erosion controlled by $|v|$, whereas isotropic measures exhibit rough beds (high values of $\xi$ and $R$) coincident with fast-flowing regions (Sects. 3.2 & 4.2). However, it is clear that fast-flow is not a



necessary condition for low roughness values (Figs. 11–13). Where ice-motion is thought not to be driven by basal sliding (in regions of slow-flow), a condition largely controlled by basal thermal state, rates of basal erosion are limited (van der Veen, 2013; MacGregor et al., 2016). It is, therefore, in these regions where we consider an alternative 'control' with regards to low $\xi$ and $R$ values (smooth beds), further elucidating characteristics of the subglacial environment.

High waveform abruptness ($A$) values, here normalised across radar sounders as $\Lambda$, have when combined with radar bed-echo reflectivity, been used to discriminate basal thermal state where larger, contiguous regions have been associated with bodies of, electrically-deep, water (Oswald and Gogineni, 2008, 2012; Oswald et al., 2018). However, recent comparison alongside ice-core temperature data and a synthesis for the likely basal thermal state (MacGregor et al., 2016) in north-west Greenland, shows this relationship to be largely inconsistent, particularly at the spatial scales (extent) assessed here (*e.g.*, Fig. 6; Jordan

et al., 2017). To build upon Jordan et al. (2017), we integrate existing knowledge of bed geology (Dawes, 2009) and information from complementary geophysical surveys (*i.e.*, gravity and magnetic anomalies; Tinto et al., 2015), to highlight that low values of $\xi$ may indeed indicate a hard-bed, particularly in large, contiguous regions ($> 1000$ km$^2$). Due to the impermeability of igneous rocks, however, low values of $\xi$ may also be a result of increased water at the ice-bed interface, giving rise to increased specularity in reflected bed-echoes (high $\Lambda$).

**4.3.1  Camp Century**

Figure 11 presents one such contiguous region of smooth bed in the vicinity of the CC drilling site; where an increase in $\xi$ is observed towards the east and south-east, near Humboldt Glacier. Fast-flowing-regions have been masked, owing to the isotropic nature of scattering-derived roughness, and the anisotropic behaviour of topographic roughness outlined above (Sect. 3.2). As the bed is likely frozen in this region (MacGregor et al., 2016), where we also observe a high elevation plateau and

slow-flowing ice (and a local ice divide), it is not feasible to interpret this signal as simply the presence of electrically-deep basal water. From the, albeit limited, knowledge of subglacial geology in this region (see Fig. 1 in Dawes, 2009), we propose that this signal (of low $\xi$) is in fact caused by a non-deformable bed, related to underlying geology on which there is little–no sediment. This bed, reminiscent of pre-glacial erosion surfaces observed in Antarctica (Rose et al., 2015), also is likely to have been largely untouched by long-term glacial erosion.

Also observed in this region are elevated $\xi$ values coincident with the Hiawatha impact crater (Kjær et al., 2018), associated with channelised features (triangle; Fig. 11). Whilst higher values of $\xi$ may well be due to the interference from off-nadir echoes (as explained above; see Sect. 4.4), it is plausible that, by contrast, this may be a marker for a soft-bed (*i.e.*, presence of deformable sediment), as a result of enhanced sediment transport.

**4.3.2  Igneous intrusion, Petermann Glacier**

Figure 12 depicts scattering-derived roughness and bed elevation near Petermann Glacier, north-west Greenland. East of the streaming ice and bounded to the north and east by the palaeofluvial 'mega-canyon' (Bamber et al., 2013), we observe a contiguous low-$\xi$ region where surface flow speed is $< 50$ ma$^{-1}$. This signal is observed coincident with a local topographic high (with a prominence of 300 m in elevation), which, unlike the surrounding topography, is largely left unmarked or dissected by

bed channels. Previous geophysical interpretation, using both gravity and magnetic anomalies derived from OIB data (see Fig. 2 in Tinto et al., 2015), has established this unit as an intruded igneous body. The unaltered nature, and geological interpretation, of this feature further lend credibility to our interpretation of low $\xi$ values as denoting a hard bed. Additionally, recent assessment of the basal thermal state, and basal water prediction derived from RES, suggest that this region is not predomi-

nantly 'wet' (MacGregor et al., 2016; Jordan et al., 2018; Chu et al., 2018) further indicating that the interpretation of water ponding is unlikely to hold here.

### 4.3.3  Volcanic province, central Greenland

Well-constrained by exposed geology at the ice-free margins of Greenland (bounded west–east by Ìngia Isbræ and Giekie Plataeu, respectively), is the presence of a volcanic province from the Palaeogene (Fig. 13; see also Fig. 1 in Dawes, 2009);

under the inland ice in central Greenland. However, the exact extent of the presence of the underlying basaltic rocks cannot be accurately determined (Dawes, 2009). At each margin of the GrIS where $|v|$ is $< 50$ ma$^{-1}$, we see good spatial agreement between $\xi$ and the mapped volcanic province. If we are to conclude that low values of $\xi$ delineate a hard-bed, it may be possible to re-draw the boundary of the volcanic province further inland from the western margin (Fig. 13). The eastern end of this 'smooth' region is spatially correlated with elevated levels of geothermal heat, as a result of the long-term tracking of the

Iceland hot-spot, a relatively thin lithosphere, and an underplated body, discussed by Rogozhina et al. (2016) and Martos et al. (2018).

### 4.3.4  Delineating deformable/non-deformable beds

In many assessments of subglacial roughness in Antarctica, smooth beds have been associated with the presence of deformable sediment beneath fast-flowing outlet glaciers (*e.g.*, Bingham and Siegert, 2009; Rippin et al., 2014; Bingham et al., 2017)

(Sect. 4.2). It is clear from the above examples, however, that fast-flow is not a necessary condition for large, coherent regions of 'smooth' bed. Where basal conditions are not indicative of enhanced ice flow (*i.e.*, slow-flowing, cold-based regions), we suggest that low values of $\xi$ are indicative of a non-deformable, hard bed. However, this will only work well away from complex terrain (*i.e.*, regions of low relief; see Sect. 4.4) Although we reject that such contiguous signals as are evidence of ponded basal water, or indeed basal thaw, it is plausible that small-scale patches of high abruptness values (high $\Lambda$/ low $\xi$ values) could

still be interpreted this way.

If we extend our conclusion that $\xi$ may be used to demarcate underlying hard-beds, focus should then be drawn to regions where deforming basal sediment and sediment transport is likely to take place. As discussed, the majority of fast-flowing outlet regions exhibit high values of $\xi$ (Fig. 6(a), and unfiltered $R$, Fig. 5(a)), which, by $R_{\parallel}$, we interpret as exhibiting basal 'streamlining' influenced by $|v|$ (akin to that observed by Bingham et al. (2017), albeit at a different scale). This, alongside the

recent evaluation that many of Greenland's outlet glaciers are driven by the availability of basal deforming sediment (Stearns and van der Veen, 2018), suggests that high values of $\xi$ are a proxy to demarcate deformable beds.


### 4.4 Roughness scale-separation and breakdown in complex terrain

As the quantification of topographic roughness ($R$) uses a defined length scale ($L = 200$ m), the interpretable scale of subglacial roughness information, and roughness 'features,' is fixed at this order of magnitude; however, understanding the scale of information provided by scattering-derived roughness ($\xi$), and the scale-separation between both roughness measures, is likely

to be variable across the ice-sheet. Theoretically, scattering-derived roughness is sensitive to roughness information at, or between, the scale of radar wave-length (order $\sim 1$ m) and that of the Fresnel zone (order $\sim 100$ m) (Shepard and Campbell, 1999). However, as the observed spatial distribution of $\xi$ is seen to be broadly similar to that of unfiltered $R$ (Figs. 5(a) & 6(a), respectively), it may be reasonable to suggest that this measure (scattering-derived roughness) may be more appropriately interpreted as defining roughness characteristics at the larger scale.

Local topography ultimately leads to the breakdown of both subglacial roughness metrics presented here, but also likely affects the degree of scale-separation across the ice-sheet. Notably, this occurs where a large step-change is observed in bed elevation ('cliff-like' regions; *i.e.*, deep subglacial valleys/trough). Here, bed-echoes are likely to exhibit more diffuse waveform characteristics due to off-nadir echoes from the valley sides; these will present erroneously high values of $\xi$ (a 'false' rough ice-bed interface), thus adversely affecting interpretation. For this reason, it may be sensible to use quantified values of topographic

roughness to infer whether values of $\xi$ are providing useful information. For example, if coincident measures of $R$ and $\xi$ are low (topographically smooth), and high, respectively, it may be clear that the subglacial environment is exhibiting more fine-scale roughness information.

Additionally, it is important to note that measures of $R$ also breakdown in similarly complex terrain, where cliff-like changes in along-track bed topography fall within the sampling window. An example of this is illustrated by the transparent grey bar

on Fig. 2. Both metrics, however, assume a Gaussian distribution about a mean surface; where local topography exhibits such step-changes it appears that this statistical model for roughness no longer holds. As such, the main conclusion we draw in this study with regard to $R$ and ice-sheet-motion remains unaffected, as the exponential scaling relationship (Figs. 7(d)) holds for the lower-end of $R_{\parallel}$ values ($R_{\parallel} \leq 10^{1.25}$), accounting for the vast majority of calculated values in fast-flowing regions.

### 5  Summary and Conclusions

We have presented the first systematic approach to quantifying subglacial roughness across the GrIS using two independent methods: statistical analysis of topography, and the properties of the bed-echo waveform/scattering. This not only provides an updated 'map' for the spatial distribution of subglacial roughness characteristics in Greenland (*cf.* Layberry and Bamber, 2001; Rippin, 2013), but, further quantifies the relationship between roughness and ice-sheet-motion. The study also helps to elucidate other spatially-heterogenous aspects of the subglacial environment. For our measure of topographic roughness

($R$), we have provided near-complete spatial coverage, making use of data from all publicly-available CReSIS radar sounding campaigns (1993–2016). Filtering $R$ with respect to surface ice velocity (*i.e.*, speed and direction) has enabled the assessment of roughness anisotropy both at the ice-sheet scale and more-locally in certain regions and at specific outlet glaciers.




Values for subglacial roughness, quantified here using both topographic- and scattering-derived metrics, suggest that the majority of fast-flowing outlet glaciers are underlain by rough beds. Conversely, the slow-flowing interior is smooth. This suggests that enhanced glacier flow (*i.e.*, basal sliding) in Greenland is either unlikely to be controlled by basal traction, following a Weertman-style hard-bed sliding parametrisation (Weertman, 1957), or rather basal traction is not induced by

the wavelengths of roughness information quantified in this study. It is clear, however, that there is pronounced anisotropy in topographic roughness with respect to ice flow direction, in fast-flowing regions ($|v| > 50$ ma$^{-1}$). Hence, topographic roughness exhibits an exponential scaling relationship with ice surface speed parallel, but not perpendicular, to flow direction. At the length scale used to calculate topographic roughness (*i.e.*, 200 m), the observed anisotropy and scaling relationships observed are likely due to enhanced rates of subglacial erosion resulting in a streamlining of bed features, possibly through

deforming basal sediment (*e.g.*, MSGLs observed in King et al., 2009; Schroeder et al., 2014; Bingham et al., 2017). We, therefore, suggest that consideration of roughness anisotropy is required with a view to infer relationships with ice-motion and subglacial processes. Additionally, in many slow-flowing regions, we conclude that contiguous smooth regions of the bed is likely due to the presence of a hard bed, rather than the presence of soft deformable sediment. In this vein, our study provides scope for a spatially variable soft-bed/hard-bed (deformable/non-deformable) boundary constraint for ice-sheet models.

*Data availability.* The two subglacial roughness metrics presented here are available for download from the Polar Data Centre, Natural Environmental Research Council. UK doi:10.5285/6071926f-32e0-4681-a50d-aab08f42c08a; URL: http://doi.org/ckqg. The L1B and L2 RES data are available from CReSIS at https://data.cresis.ku.edu/data/rds/ (last access: September 2018) and are documented in Paden (2017). The Greenland basal thermal state synthesis (MacGregor et al., 2016), ice thickness and topography data sets (BedMachine, V3) (Morlighem et al., 2017), and ice surface speed (Joughin et al., 2016), are archived by NSIDC at https://doi.org/10.5067/R4MWDWWUWQF9, https://nsidc.org/data/idbmg4

(last access: September 2018) and https://nsidc.org/data/ NSIDC-0670/versions/1 (last access: September 2018), respectively.

*Competing interests.* JLB is the advisory editor of The Cryosphere. The authors declare that they have no conflict of interests.

*Author contributions.* AA DKD

*Acknowledgements.* MAC was supported by the UK NERC grant NE/L002434/1 as part of the NERC Great Western Four + (GW4+) Doctoral Training Partnership. TMJ, MJS, CNW, and JLB were supported by UK NERC grant NE/M000869/1 as part of the Basal Properties

of Greenland project. TMJ would like to acknowledge support from EU Horizons 2020 grant 747336-BRISRES-H2020-MSCA-IF-2016. DMS was supported by a grant from the NASA Cryospheric Sciences Program. We would like to thank John Paden, CReSIS, for his advice on radar data processing, and we acknowledge the use of data products from CReSIS generated with support from NASA grant NNX16AH54G. Additionally, we thank Rob Bingham for useful comments on an earlier version of this manuscript.





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





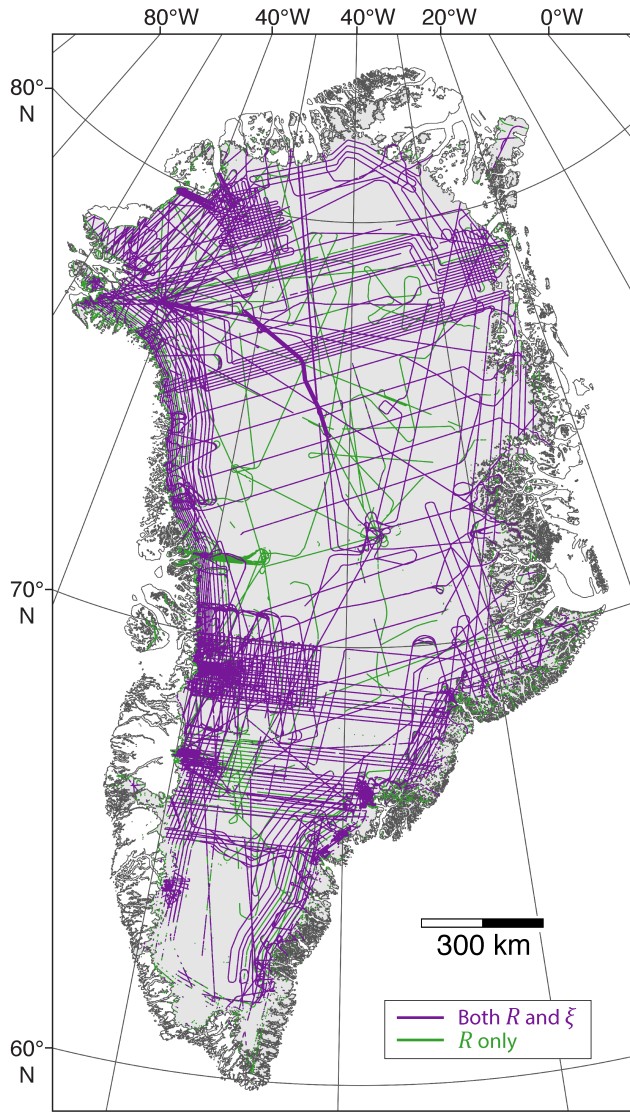

**Figure 1.** Coverage of radar sounding surveys over the GrIS used in this study. Topography-derived (topographic) roughness ($R$) is calculated using all available CReSIS survey data between 1993–2016, where scattering-derived roughness ($\xi$) uses only a subset of these (further explained in section 2.3). Displayed using a polar stereographic north projection (71° N, 39° W), as with all other spatial plots.





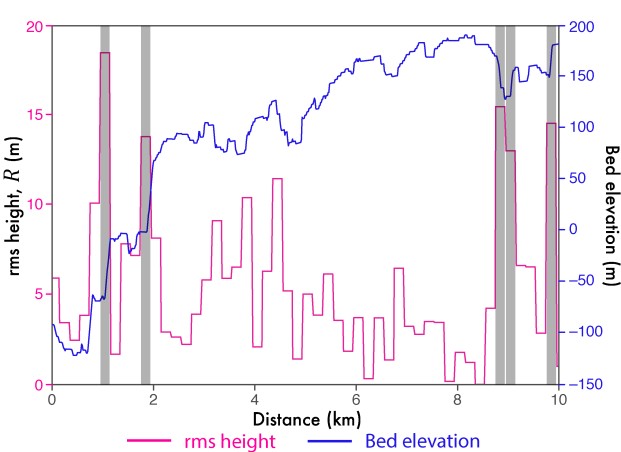

**Figure 2.** Along-track example of calculated topographic roughness ($R$). This demonstrates the length scale (200 m) over which $R$ is calculated from sampled bed elevation. Grey bars depict high values of $R$ assiociated with subglacial step-changes in elevation (cliffs); the limitation of interpreting topographic roughness in these regimes is discussed in section 4.4).



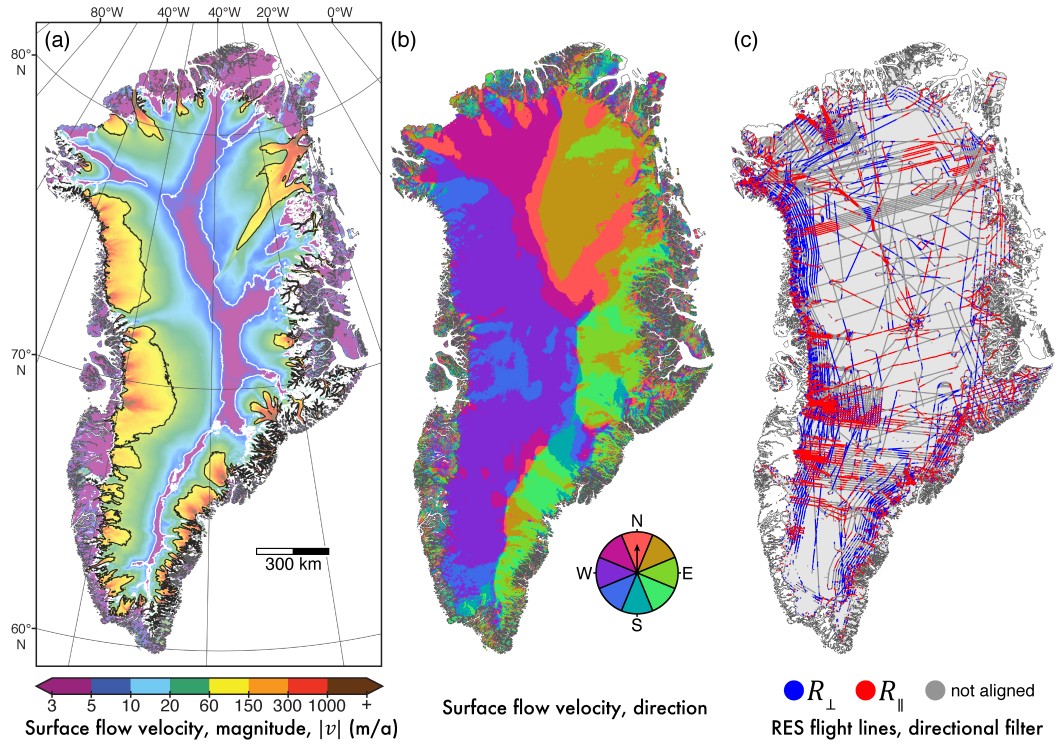

**Figure 3.** Observed surface ice velocity characteristics of the GrIS used in the filtering of $R$. (a) InSAR-derived ice surface speed (velocity magnitude; m a$^{-1}$) (Joughin et al., 2016); regions of fast ($|v| > 50$ ma$^{-1}$) and slow ($|v| < 5$ ma$^{-1}$) flow are demarcated by the black and white contour lines, respectively. (b) Flow direction of ice surface, from (a); coloured pin-wheel denotes direction of surface ice flow, where north is at the top of the page. (c) Radar sounding surveys as in Fig. 1 filtered for alignment with surface flow direction (b); flight tracks are categorised as aligned either parallel ($R_\parallel$) or perpendicular ($R_\perp$) to surface flow direction (with a 20° threshold) for the analysis of topographic roughness anisotropy.





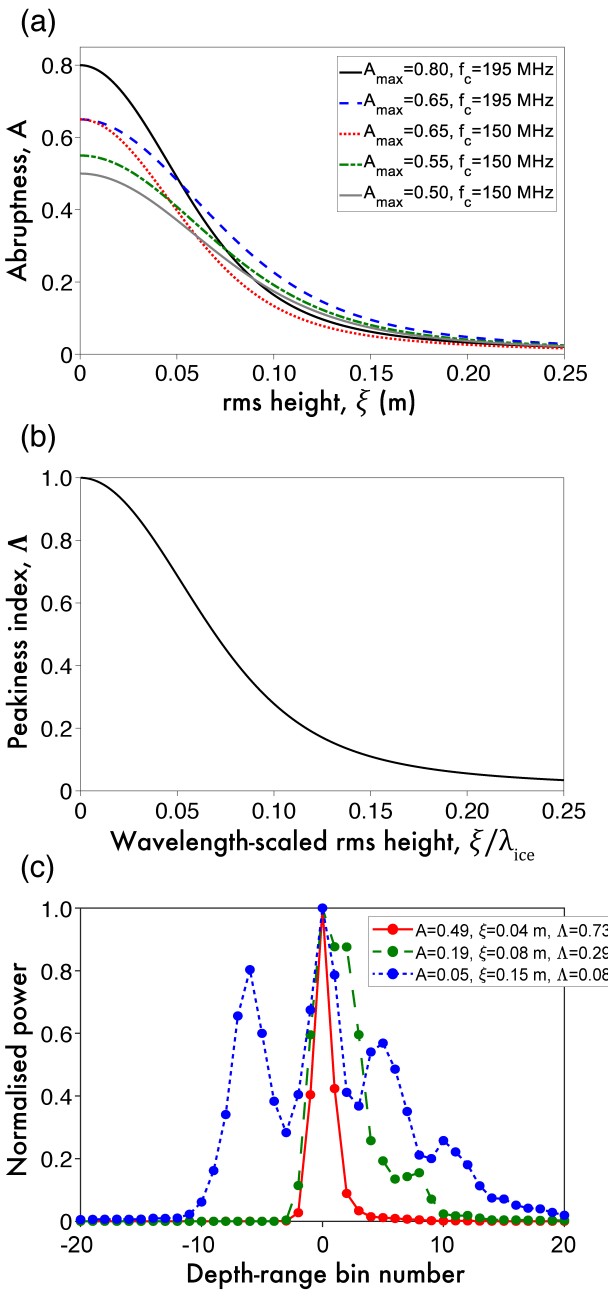

**Figure 4.** Estimation of scattering-derived roughness and data combination for bed-echo peakiness. (a) Abruptness as a function of rms height for different CReSIS field seasons: solid black curve, 2010 DC8; long dashed blue curve, 2011 TO, 2011 P3, 2012 P3, 2013 P3 and 2014 P3; dotted red curve, 2006 TO; tray-dashed green curve, 2005 TO; solid grey curve, 2008 TO and 2009 TO. (b) Peakiness Index as a function of wavelength-scaled rms height. (c) Example bed-echo waveforms, their abruptness, $A$, peakiness-index, $\Lambda$, and scattering-derived roughness, $\xi$. The plots are for the 2011 P3 field season which has maximum A $\sim$0.65.





**Figure 5.** Topographic roughness ($R$) across the GrIS. (a) $R$ unfiltered by flow direction. (b) $R_\perp$. (c) $R_\parallel$. (d) and (e) shows spatial inter-polation of (b) and (c) to a width of 20 km, respectively, for improved visualisation. Locations for Ìngia Isbræ (II) and Camp Century (CC) drilling site are marked.



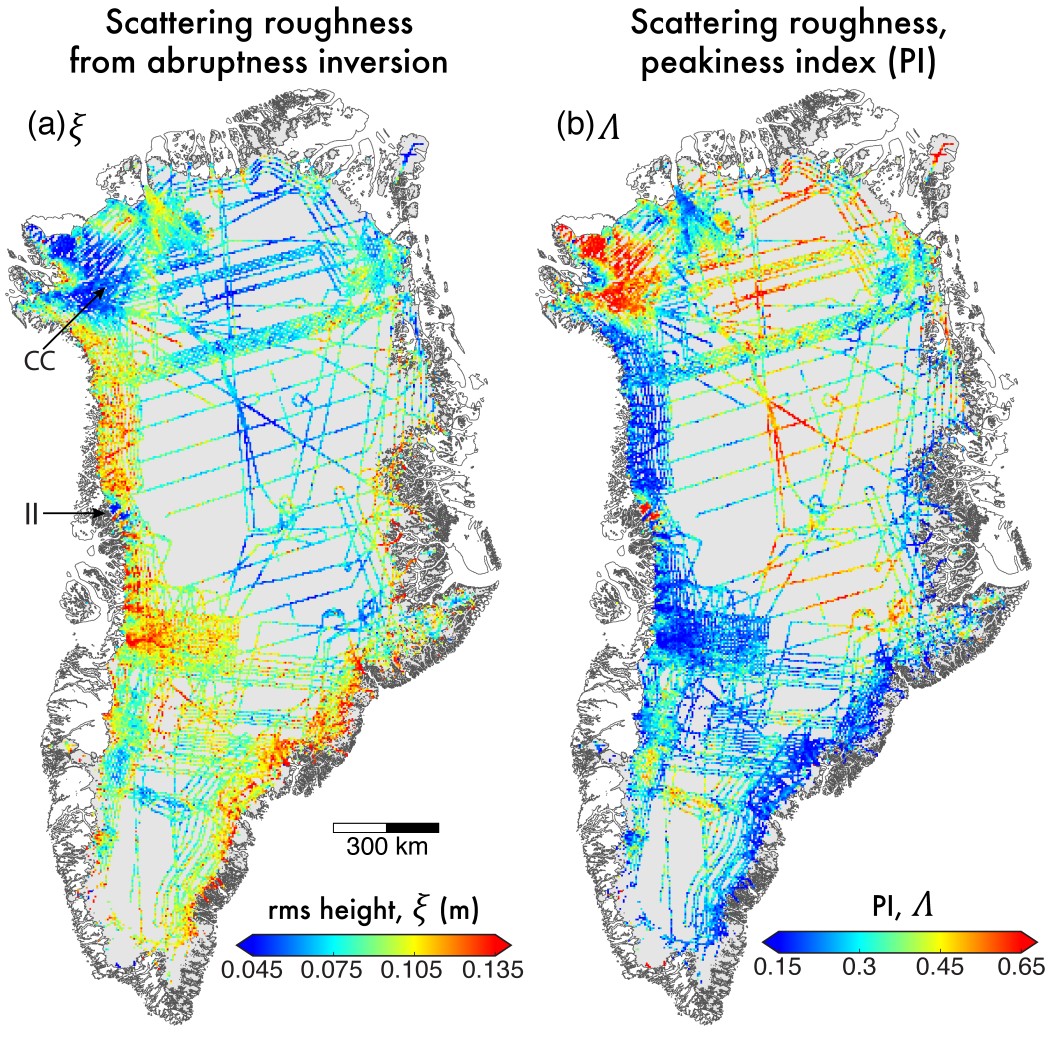

**Figure 6.** Scattering-derived roughness ($\xi$) across the GrIS. (a) $\xi$. (b) Non-dimensional $\Lambda$ (peakiness index) determined from the bed-echo waveform. Locations for Ìngia Isbræ (II) and Camp Century (CC) drilling site are marked.





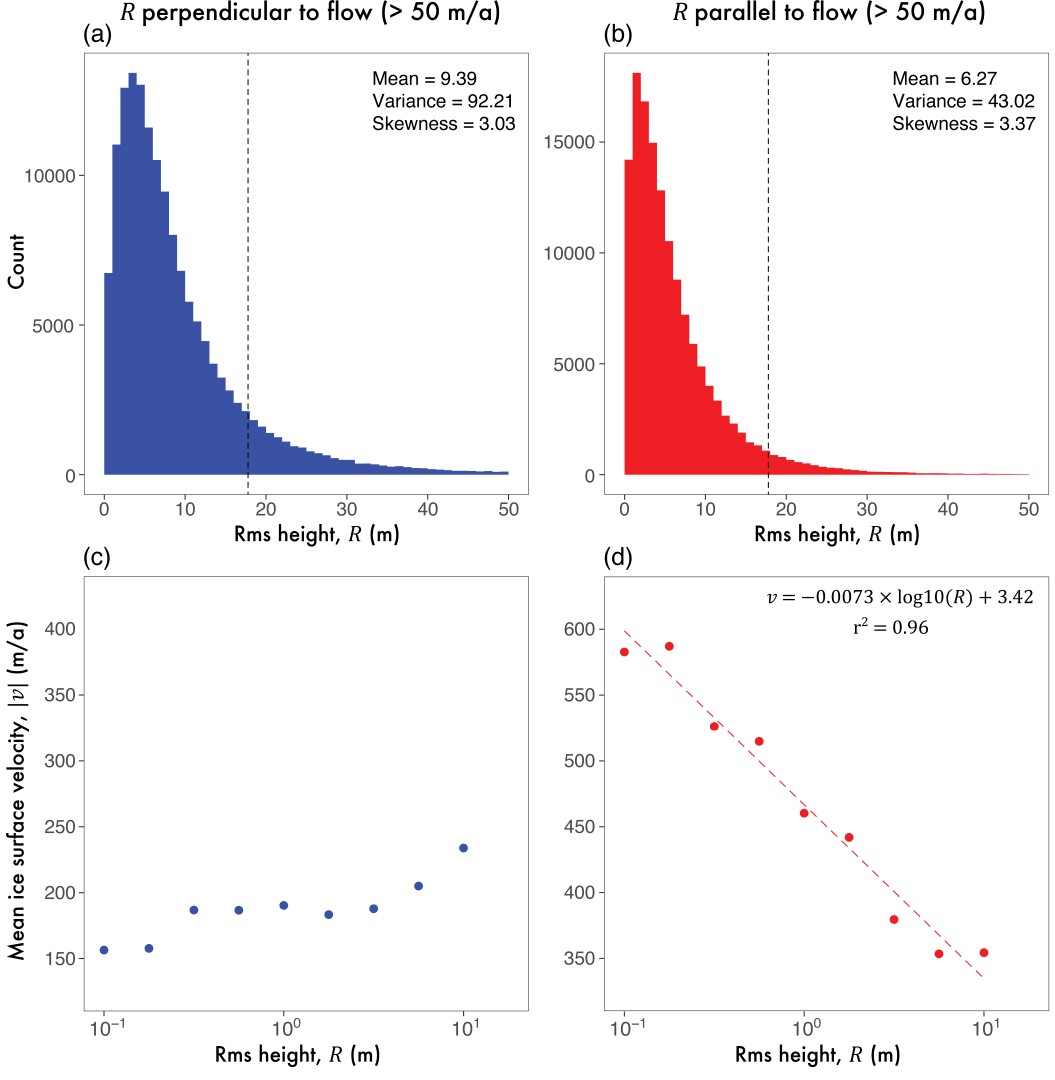

**Figure 7.** Relationship between $R_\perp$ and $R_\parallel$ and surface ice velocity for fast-flowing ($|v| > 50\,\mathrm{ma}^{-1}$) regions of the GrIS. (a) and (b) present distributions $R_\perp$ and $R_\parallel$, respectively. (c) and (d) show mean ice surface speed, $|v|$, calculations for logarithmic $R$ bins (at 0.25 m intervals). This is a linear–log plot, where the limit of the horizontal axis ($R$) is $10^{1.25}$ m, noted by the dashed black lines in (a) and (b). It should be noted the vertical exaggeration of these two plots are constant. Colours here are consistent with Fig 3 (c) for alignment with surface flow direction.



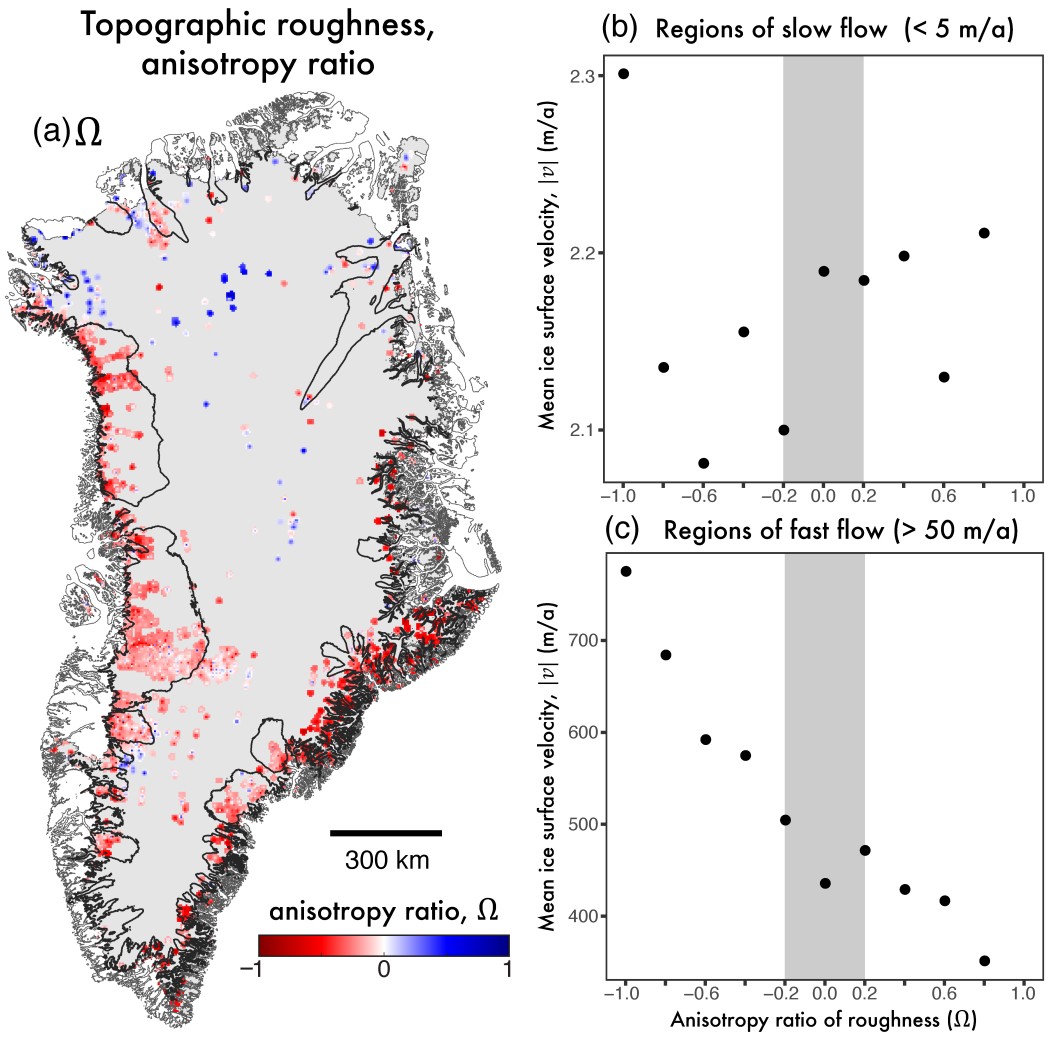

**Figure 8.** Calculated anisotropy ratio for $R$. (a) Anisotropy ratio, $\Omega$, where values of $-1$ dictate a dominance of smoothness parallel to flow direction, $+1$ a dominance of smoothness perpendicular to flow (*i.e.*, parallel roughness), and values of 0 indicate isotropy. (b) and (c) present mean ice surface speed, $|v|$, calculated for anisotropy ratio bins (at 0.1 intervals) for slow- and fast-flowing regions, respectively.



**Figure 9.** Local subsets of $R_\perp$ and $R_\parallel$ in fast-flowing outlet glacier regions. Interpolated $R_\perp$ and $R_\parallel$ (as Fig. 5) is shown for the fast-flowing regions of: (a) Humboldt [1] and Petermann [2] glaciers; (b) the North East Greenland Ice Stream (NEGIS) [3]; (c) the North West (NW) [7] fast-flow region; (d) Kangerdlugssuaq [4]; (e) Jakobshavn Isbrae and surrounding glaciers [6]; and (f) Helheim [5]. The location of these regions is inset, where regions are colour-coded for further analysis (see Figure 10).





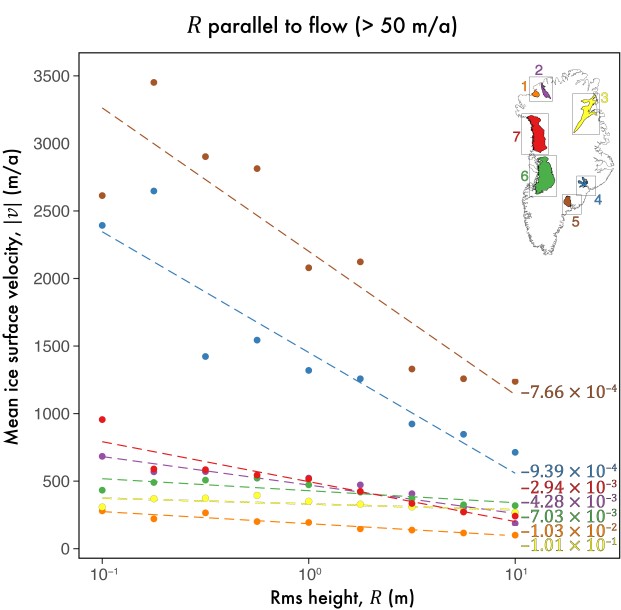

**Figure 10.** Relationship between $R_\parallel$ and ice surface speed $|v|$ for fast-flowing outlet glacier regions. This is a linear–log plot as per Figure 7 (d), depicting the calculated mean ice surface speed, $|v|$, for logarithmic $R$ bins (at 0.25 m intervals) at each of the 7 regions shown in 9; the gradient of the linear model (with units of $\mathrm{a}^{-1}$) for each region is shown in ascending order (less negative).



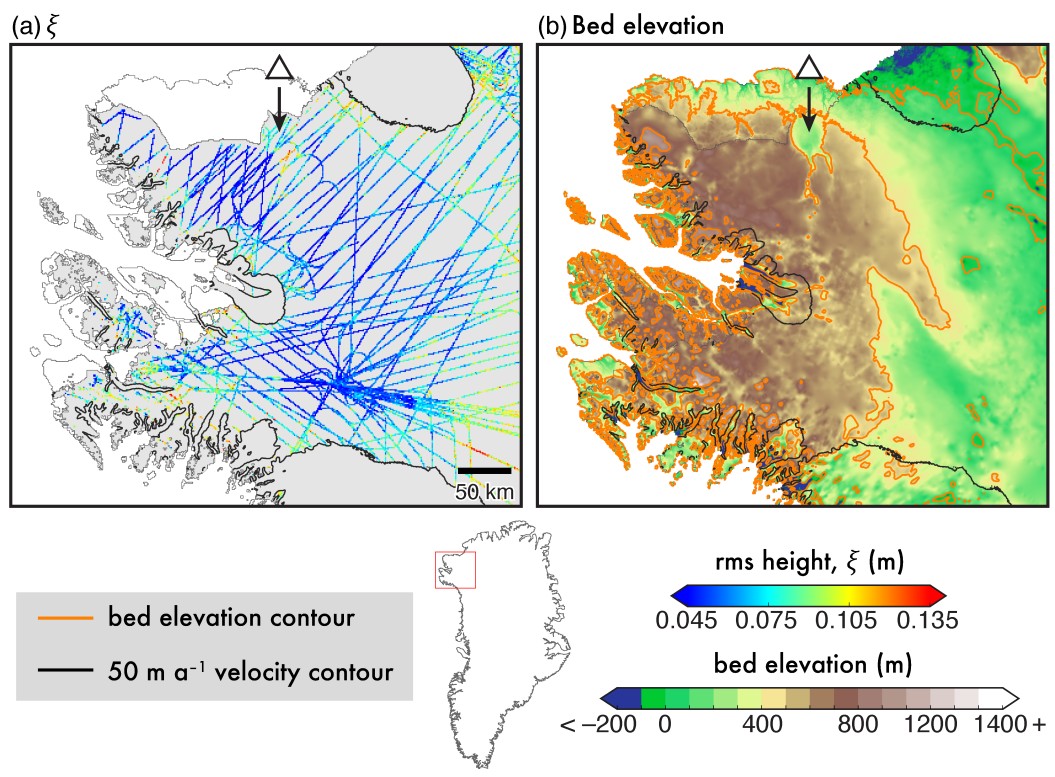

**Figure 11.** Geological interpretation using scatting-derived roughness, $\xi$, near Camp Century. (a) $\xi$, with values in fast-flowing regions (delineated by black contour; $|v| > 50$ ma$^{-1}$) masked. (b) Bed elevation (BedMachine, v3; Morlighem et al., 2017) with contours at 400 m intervals. The site of the Hiawatha impact crater (Kjær et al., 2018), associated with channelised features is marked (triangle; discussed in Sect. 4.3). Location inset.





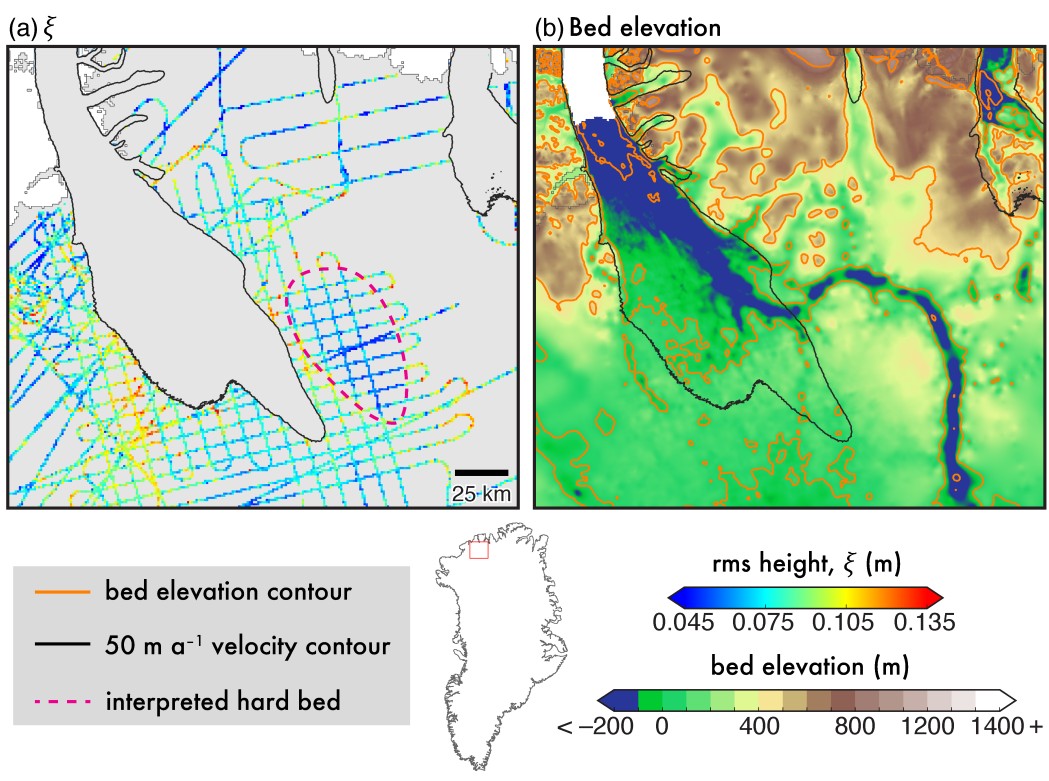

**Figure 12.** Geological interpretation using scatting-derived roughness, $\xi$, at Petermann Glacier. (a) $\xi$, with values in fast-flowing regions (delineated by black contour; $|v| > 50$ ma$^{-1}$) masked. Interpreted smooth, hard bed delineated by pink dashed line. (b) Bed elevation (BedMachine, v3 Morlighem et al., 2017)) with contours at 400 m intervals. Location inset.



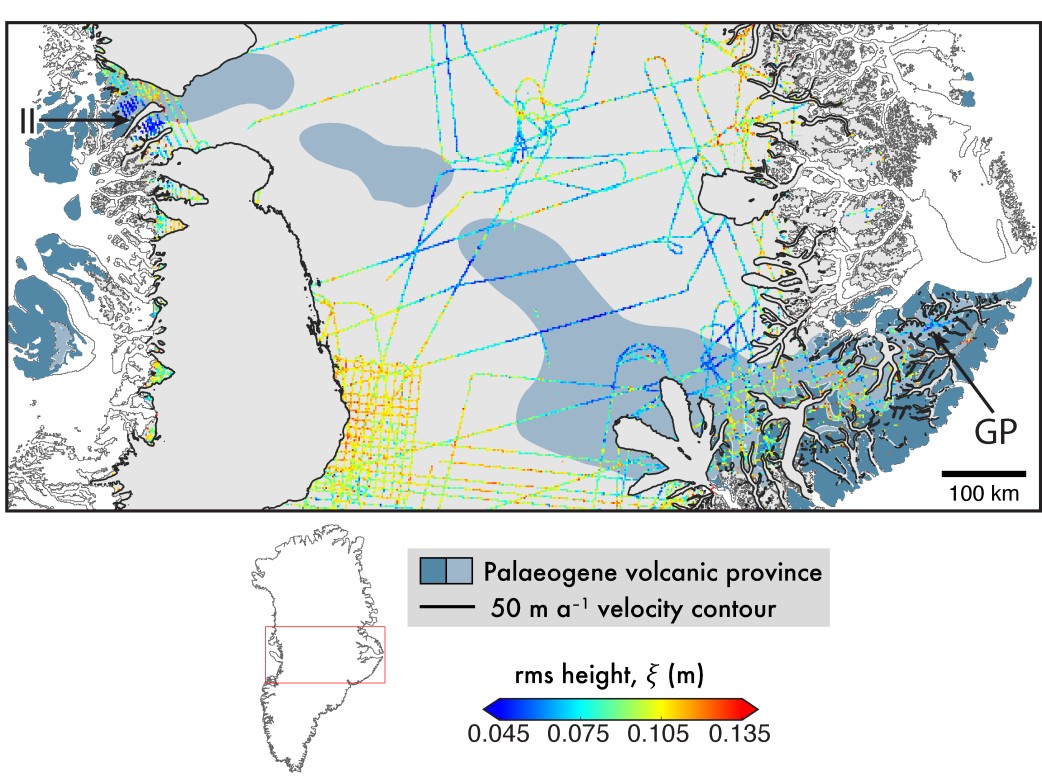

**Figure 13.** Geological interpretation using $\xi$ in central Greenland. Values of $\xi$ in fast-flowing regions are masked (delineated by black contour; $|v| > 50$ ma$^{-1}$). Exposed or ice-free (dark shade) and predicted extent (light shade) of a Palaeogene volcanic province (Dawes, 2009) is underlain. This feature is bounded west–east by Ìngia Isbræ (II) and Geikie Plateau (GP), respectively. Location inset.