# Peer review of "Subglacial roughness of the Greenland Ice Sheet: relationship with contemporary ice velocity and geology"

_The Cryosphere, 2019_

## Referee Comment (RC1) · Poul Christoffersen (Referee) · 22 Jun 2019

Cooper et al. presents a quantitative analysis of radio-echo sounding data acquired in aero-geophysical surveys over the Greenland ice sheet by CRESIS and OperationIcebridge. The primary work is new estimates of basal roughness, calculated in different directions and using two different methods. Building on previous studies in Greenland and the Antarctic, the authors show that many fast-flowing outlet glaciers in Greenland are underlain by relatively rough beds, and that the basal interior of the Greenland ice sheet is quite flat.

When roughness is considered in different directions, the authors find basal roughness

perpendicular to the ice flow direction to scale negatively and exponentially with the speed of ice motion, whereas there is no such relationship in the perpendicular direction. They also find the degree of anisotropy between roughness in the two different directions to correlate with the speed of ice flow.

The authors interpret differences in directional roughness to stem from streamlining of the bed beneath fast flowing glaciers, suggesting they are underlain by soft basal sediment in those places, whereas the flat interior is interpreted to stem from a hard and flat crystalline bed there.

The manuscript updates previous work by Layberry and Bamber (2001) and Rippin et al (2013), who used similar, although not identical techniques and less extensive radio-echo sounding data compared to this study. While the new results are not fundamentally different from past work, they establish some interesting relationships between ice flow and patterns of roughness, while also corroborating recent findings from regional studies on a larger scale. However, the spatial scale of roughness considered here is, as in past work, not sufficiently fine to make interpretations that are directly relevant to the role of roughness in the sliding process.

The manuscript is overall well written and the data are clearly presented.

The work is robust and well explained in terms of techniques and methods, with the exception of the interpolated roughness, which is justified with the argument that it improves the visualisation. Yet, it is the interpolated roughness product that is subsequently used in the statistical analysis. Ultimately, it would have been pertinent to confirm that the statistical relationships are also found in the original non-interpolated data. If that is not possible, or if the results differ, it is important to explain why and to justify the use of interpolated data on a more technical basis.

There are few typos and the writing is mostly good. My only comment is that there are some (to me at least) odd uses of hyphen. E.g. I would say that "slow-flowing glacier" can be hyphened, whereas "a slow glacier" need not be hyphened. There are also
some informal and potentially incorrect uses of / which could be avoided. There is also an important difference between 'break down' and 'breakdown'.

The use of referencing is not always proper. For example, Rippin et al. (2006, 2011, 2013, 2014) are cited >20 times, although three of the four articles are about West Antarctica and not Greenland. When referencing regional work from the Siple Coast in West Antarctica, it would be appropriate to include at least a few references from the NSF funded work there. It may be inadvertent or accidental, but there seems to be a slight tendency to self-cite in a places where it would be pertinent and relevant to cite work by others. I also recommend including a better description of previous work which have shown or inferred the presence of soft basal sediments in Greenland (e.g. Booth et al. TC 2014; Kulessa et al. Sci Adv 2017; Hofstede et al. JGR 2018) and studies that have demonstrated potentially important sedimentary controls on ice flow there (e.g. Bougamont et al. Nat Comm, 2014).

The suggestion above may help improve the conclusions, which are not always fully justified. For example, it is strangely vague to state that "This suggests that enhanced glacier flow (i.e. , basal sliding) in Greenland is either unlikely to be controlled by basal traction, following a Weertman-style hard-bed sliding parametrisation (Weertman, 1957), or rather basal traction is not induced by the wavelengths of roughness information 5 quantified in this study." I would say the latter is correct, and that the authors are not in a position to suggest that enhanced glacier flow in Greenland is unlikely to be controlled by basal traction, whatever the mechanism. The last sentence, "provides scope for" is a really marginal conclusion, which I recommend the author remove as it has already been discussed.

Finally, I wonder whether it would be appropriate to include someone in the CRESIS team as a co-author, even if it is not a requirement.

---

## Referee Comment (RC2) · Anonymous Referee #2 · 19 Jul 2019

This study estimates the subglacial roughness under the Greenland Ice Sheet by two methods (along track topography and radar scattering) using comprehensive radar data sets. While not new in the methods and results, this study complete previous studies by presenting up-to-date results for the whole GrIS. The roughness is compared to the observed surface velocities and to other geophysical informations.

The methods and data are clearly described. My main reserve is about the interpretation of the roughness in terms of sliding law and processes controlling the basal friction. The fact that the interior (slow flow regions) appears to be smoother than the margins (fast flow regions) is used to invalidate the applicability of the Weertman law to model

the basal friction conditions under the GrIS (Section 4.1, Page 13, Lines 4-16 and Section 5, Page 18, Lines 2-5). This discussion is rather hypothetic as, as you mention several times (e.g. page 14 lines 10-20), Weertman theory is based on the influence of the small scale rugosity (centimeters to meters) while you measure the topographic rugosity (at least with $R$). Moreover, in Weertman theory the sliding speed is function of the friction coefficient (depending on the rugosity, higher rugosity leading to higher friction coefficient) and of the basal stress. As the basal stress vary from place to place, it is not possible to draw conlusions on the influence of the rugosity using only the velocities. This would require to use an ice flow model to estimate the basal stress and correlate the rugosity with the effective friction coefficient.

From the abstract and more clearly page 14 lines 10-20, we understand that it is better to interpret the relation between the topographic rugosity and the velocity, in terms of erosion processes, and so the effect of the velocity on the topography. I think this should be clarified and the interpretations in terms of rugosity affecting the velocity should be let aside.

Also, I think it would be more clear if you give more details about what is known from the subglacial geology under the GrIS in the Introduction (e.g. introducing the volcanic province, the known igneous intrusions and other related geological informations).

**1 Minor comments**

- Page 1, Line 18: *primarily driven by mass loss over the grounding line [...]*; I think this should be rephrased to reflect the fact that the mass loss of the Greenland ice sheet is partitioned between increased ice discharge and increased surface melt. The exact numbers for the contribution of each component depends on the studies and time periods. It could be useful to include references to the most recent studies.

- Page 2, Line 22: reference to *Durand et al., 2011* is not appropriate in this context. Better to cite Gillet-Chaulet et al., The Cryosphere, 2012 for the inversion of the basal conditions under the GrIS.

- Page , Line 5: *the causes and controls smooth-and rough-beds [...].* "of" missing between controls and smooth?

- Page 4, Line 6 and 7: *higher abruptness and associated.* "and" should be "is"? idem latter in the sentence *lower abruptness "is" associated with fine scale [...]*

- Page 6, Line 8: *to insure only independent measures of bed elevation were used.* Could you explain this?

- Page 6, Lines 17-24: Maybe you could illustrate the influence of $L$ in Fig. 2, to show how sensitive are the results to this value?

- Page 6, Line 18: *for a length scale not less than 100m.* Could you explain this value of 100m for the lower bound.

- Page 6, Lines 23-24: *This repeated sampling approach for small $n$ [...].* I don't understand the meaning of this sentence.

- Section 2.2.2: I think you should include a discussion on the uncertainty on the flow direction especially for the 'slow' flow regions.

- Section 2.3.2: explain how $A_{max}$ depends on the radar system.

- Page 9, Line 10: *[...] and then re-scaled amplitude on the interval [0,1].* Explain the rescaling, Is it $A/A_{max}$?

- Figure 4: (a) and (b) x-axes have different units ($\xi$ and $\xi/\lambda$) but same values (between 0 and 0.25), is this correct?

- Section Results and associated Figures; Please when you discuss specific areas in the text (e.g. Petermann, Humbolt, NEGIS, Camp Century, etc...) mare sure that the names are given in the corresponding figures, or include a figure with names of the places that are cited in the manuscript

- Section 3.2.1 and Figure 7: The fact that the mean velocity do not exceed 250 m a$^{-1}$ for $R_\perp$ but is $> 350$ m a$^{-1}$ for $R_\parallel$ is only possible because the spread of the velocity is larger in the bins for $R_\perp$? I think it could be interesting to use box and whiskers in Fig. 7 to discuss this? Idem for Fig. 8.

- End of Section 3.2.1, discussion on the anisotropy for slow flow regions: Could this be due to the uncertainty in the flow directions (cf comment above)?

- Section 3.2.2, Page 11, Line 30: $R_\perp$ should be $R_\parallel$.

- Page 12, Line 1: *Regionally, [...]*: include a reference to Fig. 10.

- Page 13, last paragraph: you seem to suggest that the smoother interior could have been produced by the waxing and waning of the ice sheet, however your results suggest that fast flow at the margin produces rougher bed, is this not a contradiction?

---

## Author Comment (AC1) · 24 Aug 2019

**Response to reviewers: Subglacial roughness of the Greenland Ice Sheet: relationship with contemporary ice velocity and geology**

**Michael Cooper, University of York (formerly University of Bristol)**

We would like to thank the reviewers for their detailed and constructive comments and feedback. We are also grateful for the opportunity to present an improved, revised version of our manuscript for re-submission. Below are our responses (in red) to both sets of received reviewer comments (italicised, black).

**One comment raised by both reviewers highlights the, perhaps, over-interpretation of calculated roughness metrics with reference to the applicability (or invalidity) of the Weertman law in Greenland. Poul Christoffersen (reviewer #1), notes:**

- *However, the spatial scale of roughness considered here is, as in past work, not sufficiently fine to make interpretations that are directly relevant to the role of roughness in the sliding process.*
- *…For example, it is strangely vague to state that "This suggests that enhanced glacier flow (i.e., basal sliding) in Greenland is either unlikely to be controlled by basal traction, following a Weertman-style hard-bed sliding parametrisation (Weertman, 1957), or rather basal traction is not induced by the wavelengths of roughness information quantified in this study." I would say the latter is correct, and that the authors are not in a position to suggest that enhanced glacier flow in Greenland is unlikely to be controlled by basal traction, whatever the mechanism.*

**And reviewer 2 elaborates:**

- *My main reserve is about the interpretation of the roughness in terms of sliding law and processes controlling the basal friction. The fact that the interior (slow flow regions) appears to be smoother than the margins (fast flow regions) is used to invalidate the applicability of the Weertman law to model the basal friction conditions under the GrIS (Section 4.1, Page 13, Lines 4-16 and Section 5, Page 18, Lines 2-5). This discussion is rather hypothetic as, as you mention several times (e.g. page 14 lines 10-20), Weertman theory is based on the influence of the small scale rugosity (centimeters to meters) while you measure the topographic rugosity (at least with R). Moreover, in Weertman theory the sliding speed is function of the friction coefficient (depending on the rugosity, higher rugosity leading to higher friction coefficient) and of the basal stress. As the basal stress vary from place to place, it is not possible to draw conclusions on the influence of the rugosity using only the velocities. This would require to use an ice flow model to estimate the basal stress and correlate the rugosity with the effective friction coefficient.*
- *…From the abstract and more clearly page 14 lines 10-20, we understand that it is better to interpret the relation between the topographic rugosity and the velocity, in terms of erosion processes, and so the effect of the velocity on the topography. I think this should be clarified and the interpretations in terms of rugosity affecting the velocity should be let aside.*

**A response to both reviewers, regarding these points:**
We agree that the evaluated length-scale for R (our topographic roughness measure) does not allow us to be definitive in concluding the invalidity of the Weertman sliding law in Greenland. However, and as reviewer #2 suggests, the scattering derived roughness metric, which has some sensitivity at the wavelength of the radar (~1 metre), will indeed capture

some information at a scale comparable to that of influential small-scale rugosity (and depicts 'rough' margins). Furthermore, with respect to reviewer #2's comment, we agree that in principle, roughness data could have been compared directly with a calculated/ modelled friction coefficient output in this study (as in Bingham et al., 2017 (https://doi.org/10.1038/s41467-017-01597-y); however, as large-scale structure of surface velocity inversely correlates with beta (the friction coefficient) (see Perego et al., 2014: https://doi.org/10.1002/2014JF003181), we decided to use ice surface velocity as it is directly observable.

Regardless, we do agree that, in the original manuscript, the phrasing of our discussion and conclusions were too definitive. As such, we have adjusted the manuscript to better state that the length-scale over which topographic roughness (R) is evaluated does not allow direct inference regarding Weertman-style sliding laws, and even where scattering-derived roughness may provide some information in this regard, it is inappropriate to parameterise bed friction in a general way using these metrics. Please refer to the following lines/ paragraphs with respect to these changes:

- Section 4.1, Page 14, Lines 20—31;
- Summary and Conclusions (Section 5), Pages 19—20, Lines 31—2;
- with more minor changes throughout.

**Responses to general, reviewer-specific, comments:**

**Reviewer #1 (Poul Christoffersen):**
*The work is robust and well explained in terms of techniques and methods, with the exception of the interpolated roughness, which is justified with the argument that it improves the visualisation. Yet, it is the interpolated roughness product that is subsequently used in the statistical analysis. Ultimately, it would have been pertinent to confirm that the statistical relationships are also found in the original non-interpolated data. If that is not possible, or if the results differ, it is important to explain why and to justify the use of interpolated data on a more technical basis.*
To clarify, it is in fact only the original, non-interpolated data that are used within the statistical analysis presented in the paper, whereby the interpolated roughness values are only used for visualisation purposes (in parts d & e of Figure 5). We believe that it was unclear phraseology in the original manuscript which brought about this confusion; in the revised manuscript we have sought to improve clarity, and to avoid further confusion, with minor changes made in Sections 3.1.1 (Page 11, Line 16) and 3.2 (Page 12, Line 15).

*There are few typos and the writing is mostly good. My only comment is that there are some (to me at least) odd uses of hyphen. E.g. I would say that "slow-flowing glacier" can be hyphened, whereas "a slow glacier" need not be hyphened. There are also some informal and potentially incorrect uses of / which could be avoided. There is also an important difference between 'break down' and 'breakdown'.*
With respect to hyphens, general spelling and grammar, and the use of slashes (/), various changes have been made throughout the manuscript.

*The use of referencing is not always proper. For example, Rippin et al. (2006, 2011, 2013, 2014) are cited >20 times, although three of the four articles are about West Antarctica and not Greenland.*

*When referencing regional work from the Siple Coast in West Antarctica, it would be appropriate to include at least a few references from the NSF funded work there. It may be inadvertent or accidental, but there seems to be a slight tendency to self-cite in a places where it would be pertinent and relevant to cite work by others. I also recommend including a better description of previous work which have shown or inferred the presence of soft basal sediments in Greenland (e.g. Booth et al. TC 2014; Kulessa et al. Sci Adv 2017; Hofstede et al. JGR 2018) and studies that have demonstrated potentially important sedimentary controls on ice flow there (e.g. Bougamont et al. Nat Comm, 2014). The suggestion above may help improve the conclusions, which are not always fully justified.*

As above, with respect to the general use of referencing and/or citation, various changes have been made throughout the manuscript in order to correct any misuse, or improper referencing.

Additionally, more substantial changes have been made to the manuscript in order to better reference and describe previous work:
- the inclusion of introductory sentences (and where relevant, changes to the discussion, Sect. 4.3.4) regarding the presence of soft basal sediments, and their controls on ice flow, in Greenland (Introduction, Page 3, Lines 10—21);
- and, the inclusion of references to the work undertaken within the Siple Coast, West Antarctica where relevant, both in the introduction and the discussion (Page 18).

*The last sentence, "provides scope for" is a really marginal conclusion, which I recommend the author remove as it has already been discussed.*

Removed as suggested.

*Finally, I wonder whether it would be appropriate to include someone in the CRESIS team as a co-author, even if it is not a requirement.*

A CReSIS team member was included in a prior publication (an integral pre-cursor to this work, Jordan et al., 2017: https://doi.org/10.5194/tc-11-1247-2017); that publication dealt with the original extraction of radar power and waveforms from the bed picks (which were then used in this manuscript), requiring more direct collaboration with the data collection team there.

**Reviewer #2:**
*…I think it would be more clear if you give more details about what is known from the subglacial geology under the GrIS in the Introduction (e.g. introducing the volcanic province, the known igneous intrusions and other related geological information).*

Following this comment, we have now included some introductory sentences regarding related geological information. Please see Page 3, Lines 28—end.

**Minor comments**
*• Page 1, Line 18: primarily driven by mass loss over the grounding line […]; I think this should be rephrased to reflect the fact that the mass loss of the Greenland ice sheet is partitioned between increased ice discharge and increased surface melt. The exact numbers for the contribution of each component depends on the studies and time periods. It could be useful to include references to the most recent studies.*

Agreed, the manuscript has been adjusted accordingly. Please see Pages 1—2

*• Page 2, Line 22: reference to Durand et al., 2011 is not appropriate in this context. Better to cite Gillet-Chaulet et al., The Cryosphere, 2012 for the inversion of the basal conditions under the GrIS.*

Done

• *Page , Line 5: the causes and controls smooth-and rough-beds […]. "of" missing between controls and smooth?*

Yes, fixed.

• *Page 4, Line 6 and 7: higher abruptness and associated. "and" should be "is"? idem latter in the sentence lower abruptness "is" associated with fine scale […]*

Fixed.

• *Page 6, Line 8: to insure only independent measures of bed elevation were used. Could you explain this?*

The along-track sample spacing of the more recent data is approximately twice the horizontal resolution that occurs from SAR processing followed by multi-looking – the manuscript has been adjusted to clarify this (See page 6, lines 30—end).

• *Page 6, Lines 17-24: Maybe you could illustrate the influence of L in Fig. 2, to show how sensitive are the results to this value?*

We have now included reference to our previous paper, an important pre-cursor for this research (see, above; Jordan et al 2017), and to works of Shepard (1995, 1999) which better explain the statistical behaviour of subglacial terrain. These suggest that subglacial terrain exhibits self-affine (fractal) scaling behaviour; therefore, as L increases, regions with steeper slope (greater Hurst exponent) will tend to become more rough (relatively) to regions with lower slope. Please see page 7, lines 20—23.

• *Page 6, Line 18: for a length scale not less than 100m. Could you explain this value of 100m for the lower bound.*

This is due to the sample spacing of the radar (now clarified in the manuscript; Pages 6—7 Lines 30—2).

• *Page 6, Lines 23-24: This repeated sampling approach for small n […]. I don't understand the meaning of this sentence.*

This sentence has now been removed as it was unnecessary.

• *Section 2.2.2: I think you should include a discussion on the uncertainty on the flow direction especially for the 'slow' flow regions.*

Agreed; in terms of fractional error, slow-flow regions represent the worst case scenario. The propagation of errors (both in speed and direction) has been assessed and some discussion has been added based upon this within the relevant section (Please see Page 8, Lines 15—20).

The average error in speed and direction in regions of slow flow are 0.51 m/a, and 14.55 degrees, respectively; to come to these figures, in each case, we applied the general error propagation formula for independent variables vx and vy with \theta=atan(vy/vx) and |v|=sqrt(vx^2+vy^2). Owing to the larger angular threshold within our 'alignment' classification, we believe the error in slow-flowing regions does not affect our conclusions.

• *Section 2.3.2: explain how Amax depends on the radar system.*

For MCORDS 2 data Jordan et al., 2017 demonstrated that Amax depends on the ratio of the fast-time sample spacing to the depth-range resolution. For the older data this relationship holds approximately and Amax was determined empirically from the abruptness distribution. This is now explicitly stated in the manuscript (See page 9, lines 11).

*• Page 9, Line 10: [...] and then re-scaled amplitude on the interval [0,1]. Explain the rescaling, Is it A/Amax?*
Yes, this is just a linear re-scaling using A/Amax (See page 10, lines 16—17).

*• Figure 4: (a) and (b) x-axes have different units (ξ and ξ/λ) but same values (between 0 and 0.25), is this correct?*
Yes, this is correct. The reason for the similarity is that the in-ice wavelength for MCORDS is close to a metre; this is stated this on page 9: "either 0.87 m or 1.13 m for the 195 MHz and 150 MHz systems, respectively."

*• Section Results and associated Figures; Please when you discuss specific areas in the text (e.g. Petermann, Humbolt, NEGIS, Camp Century, etc...) mare sure that the names are given in the corresponding figures, or include a figure with names of the places that are cited in the manuscript*
Done, see edits to Figures 1, 5, and 6, as well as to the Results section specifically.

*• Section 3.2.1 and Figure 7: The fact that the mean velocity do not exceed 250 m a−1 for R⊥ but is > 350 m a−1 for R∥ is only possible because the spread of the velocity is larger in the bins for R⊥? I think it could be interesting to use box and whiskers in Fig. 7 to discuss this? Idem for Fig. 8.*
Higher mean values (of ice speed) for R∥ are in fact due to the greater spread/range of values in the parallel; however, the use of the mean velocity in this paper is to ensure direct comparability to previous studies in Greenland undertaken by Lindbäck and Pettersson, 2015 (ref: https://doi.org/10.1016/j.geomorph.2015.02.027). We have added note of this to the manuscript, see page 12, lines 23—24, and to the caption of Fig. 7.

*• End of Section 3.2.1, discussion on the anisotropy for slow flow regions: Could this be due to the uncertainty in the flow directions (cf comment above)?*
See above re: error propagation.

*• Section 3.2.2, Page 11, Line 30: R⊥ should be R∥.*
Corrected

*• Page 12, Line 1: Regionally, [...]: include a reference to Fig. 10.*
Done

*• Page 13, last paragraph: you seem to suggest that the smoother interior could have been produced by the waxing and waning of the ice sheet, however your results suggest that fast flow at the margin produces rougher bed, is this not a contradiction?*
We believe that due to the unconstrained (topographically) movement (and successive waxing and waning) of the ice sheet over multiple glacial cycles would lead to a largely smooth, flat, low-lying terrain as a result of glacial scour; this is now clarified, with reference to relevant texts regarding glacial landscapes/ erosion, in the text (See Page 15, Lines 18—19.